# Interstadial diversity of East Asian summer monsoon linked to changes of the Northern Westerlies

Xiyu Dong [1], Xu Zhang [2] ✉, Haiwei Zhang [1] ✉, Yuao Zhang[3], Sune O. Rasmussen [4], Rui Zhang [1], Yanjun Cai [1], Shouyi Huang[1], Gayatri Kathayat[5], Carlos Pérez-Mejías [1], Baoyun Zong[1], Dianbing Liu[6], Pengzhen Duan[7], Anders Svensson [4], Christoph Spötl [8], Youwei Li[1], Xiaowen Niu [1], Jian Wang [1], Hanying Li[1], Youfeng Ning[1], Yao Xu[1], Xianfeng Wang [9], Nicolás M. Stríkis [10], Francisco W. Cruz [10], Ashish Sinha [1,11], Martin Werner [12], R. Lawrence Edwards[13] & Hai Cheng [1,5,14] ✉

During the last glacial period, the iconic Greenland ice-core records provide evidence of interstadial warmings with various durations ranging from a century to millennia. However, whether differences in interstadial duration are mirrored by distinct hydroclimate responses in the tropics remains unclear. Here we present four speleothem $\delta^{18}O$ records from the Indian summer monsoon (ISM) and East Asian summer monsoon (EASM) regions, spanning both short and long interstadials during the last glacial period. Greenland and ISM records show broadly similar isotopic responses across events, however, the EASM records exhibit markedly different $\delta^{18}O$ depletions between short and long interstadials. Using an isotope-enabled climate model, we attribute these differences to a further northward shift of the Northern Westerlies during short interstadials, driven by intensified high-latitude warming. This shift promoted the northwestward expansion of Western Pacific Subtropical High and hence the delivery of isotopically enriched near-sourced vapor to eastern China, dampening $\delta^{18}O$ depletion during stadial-to-interstadial transitions. Our findings highlight a previously unrecognized sensitivity of EASM precipitation $\delta^{18}O$ to nuanced meridional shifts in the Northern Westerlies in contrast to the uniform responses of the ISM during interstadials.

The last glacial period (~115–11.7 thousand years before 1950 CE, ka BP) was characterized by a series of episodic abrupt climate variability, known as Dansgaard-Oeschger (DO) oscillations[1–3], which were initially identified in Greenland ice-core $\delta^{18}O$ records. These oscillations comprise rapid transitions from cold-dry Greenland Stadials (GS) to warm-humid Greenland Interstadials (GI)[4,5]. Changes in the strength of the Atlantic Meridional Overturning Circulation (AMOC)

have been commonly used to explain the reoccurring natures of DO events[6–8], involving the northward transport of warm tropical water masses and the southward export of polar cold waters at depth (Supplementary Fig. 1), exerting far-reaching impacts beyond the North Atlantic[9–12]. High-resolution ice-core sampling further revealed that DO events were of different durations[4,5,13]. Short GIs lasted only about a century, marked by transient warming spikes, while long GIs

lasted up to several thousand years, sustaining relatively stable warm conditions[5].

DO events have been accompanied by large-scale tropical hydroclimate variations, e.g., changes in the Asian summer monsoon (ASM) system (e.g., ref. 9,14–17). The ASM system, comprising the Indian summer monsoon (ISM) and the East Asian summer monsoon (EASM) subsystems[18–20] (Fig. 1d), serves as a major source of heat and moisture, playing a critical role in inter- and intra-hemisphere climate teleconnections (e.g., refs. 9,14,21–23). Nonetheless, most process-understanding studies regarding DO events have focused on the northern and southern high latitudes and on the long DOs (e.g., refs. 24–27). This hampers a complete understanding of the impacts and associated mechanisms of the DO oscillation.

The EASM, influenced by the boreal westerly jet stream and the Tibetan Plateau (TP) orography[28–32], exhibits different hydroclimate responses compared to the ISM, which operates primarily as a tropical system driven by low-level southwesterlies[33] ("Methods"). Marine and lacustrine records have documented the fundamental pattern of northward westerly jet displacements during interstadials versus southward shifts during stadials[34,35]. However, it remains unclear how these jet movements influence moisture transport pathways and precipitation δ18O signatures at centennial- to millennial- timescales. While Chiang et al.[29] examined Westerlies-East Asian precipitation δ18O relationships, their analysis focuses on interannual timescales, excluding both Southeast China and glacial millennial-scale changes. Notably, existing studies have not systematically compared hydroclimatic responses between short and long interstadials, despite their distinct thermal characteristics – transient warming versus relatively stable

warm conditions. This represents a critical knowledge gap in understanding monsoon behavior under varying meridional temperature gradients. The current lack of high-resolution paleoclimate records, particularly for short DO events, further limits our ability to assess westerly jet influences on regional moisture sources and isotopic compositions.

To fill these knowledge gaps, we established three speleothem δ18O (δ18O$_c$) records from China, one from India, and one from Brazil, spanning DO-15.1, corresponding to GI-15.1, a centennial-scale DO event during the early Marine Isotope Stage (MIS) 3. These records are characterized by sub-centennial to centennial age precision and high resolution (5–10 years), allowing us to disentangle the detailed structure of the short-duration event (ca. 100 years) (cf. ref. 5). This, together with the comparison of longer interstadials in MIS 3, provides an opportunity to explore the hydroclimate variability of the monsoon system during interstadials of different durations in the last glacial period.

Using a common chronology, we identified a smaller isotopic amplitude in the Southeast China δ18O$_c$ records during DO-15.1 compared to that of long interstadials, which is in contrast to a comparable amplitude among events observed in Greenland ice cores and ISM δ18O$_c$ records. Applying an isotope-enabled climate model, we demonstrate that the boreal summer mid-latitude Westerlies exhibited consistent northward displacements during both short and long interstadials. Notably, the short interstadials were associated with a further northward shift, allowing more δ18O-enriched near-source moisture/water vapor (from western North Pacific) to be transported to the EASM domain and hence causing a subdued amplitude of δ18O$_c$

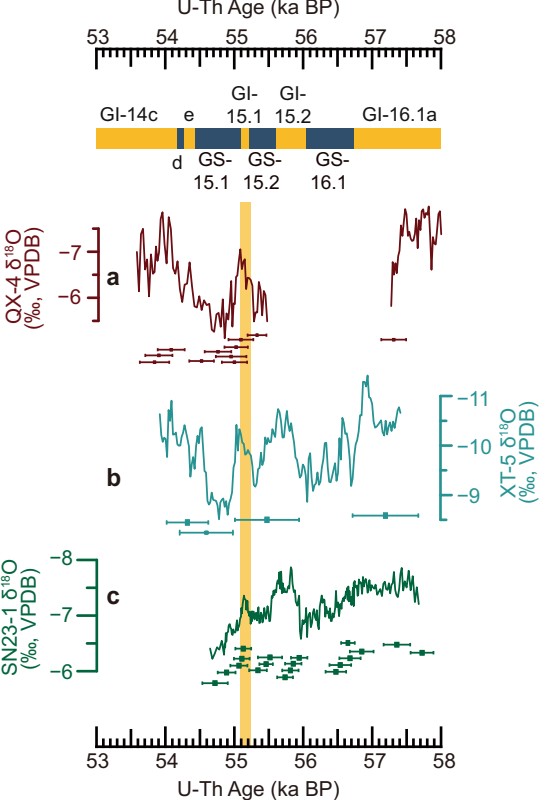

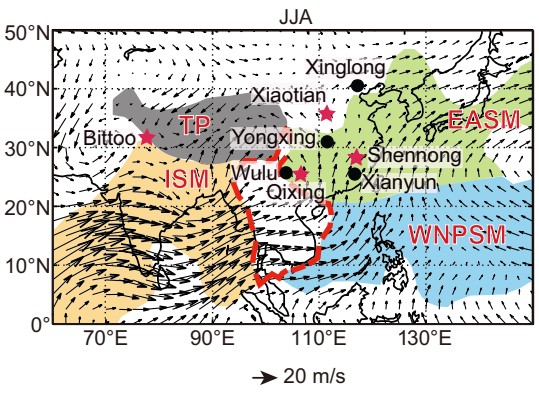

**Fig. 1 | Speleothem δ18O records. a–c** Speleothem records from Qixing cave (this study and ref. 64), Xiaotian cave (this study) and Shennong cave (this study), China, respectively. The vertical bar indicates DO-15.1. Error bars show U-Th dates with uncertainties (2σ) for each record (color coded). Cave locations are shown in (**d**), red stars show locations for the caves presented in this study, while the black dots indicate the location of other caves mentioned in this study. The black arrows are the June-July-August (JJA) monthly 850 hPa low-level wind (black arrows), data

obtained from the Global Precipitation Climatology Project (GPCP) (https://www. esrl.noaa.gov/psd/data/gridded/data.gpcp.html). The shaded areas represent the approximate locations of various monsoon systems[18], while the red dashed line outlines the approximate boundary of the transitional zone between the EASM and the ISM. *ISM* Indian summer monsoon; *EASM* East Asian summer monsoon. *WNPSM* Western North Pacific summer monsoon; *TP* Tibetan Plateau.

depletion. In contrast, regions that are dominated by tropical monsoon processes (i.e., ISM) show a muted change in water isotopic composition across long and short DO events because the moisture sources remained largely unchanged irrespective of the duration of the interstadials.

## Results and discussion

### Speleothem $\delta^{18}O$ records

In this study, five cave $\delta^{18}O_c$ records were reconstructed based on 48 U-Th dates (Supplementary Fig. 2) and ~1650 stable isotope ($\delta^{18}O$) data ("Methods"), all covering DO-15.1. Among them, three $\delta^{18}O_c$ records are from Chinese caves (Fig. 1a–c): the Shennong cave record is from the EASM domain, the Xiaotian cave record from the fringe of the EASM domain, and the Qixing cave record from the transitional zone between the EASM and the ISM (Fig. 1d); one record is from Bittoo cave in the ISM domain (Figs. 1d and 2e), and one record is from Paraíso cave in the Amazon Basin in the South American summer monsoon (SASM) domain (Fig. 2g and Supplementary Fig. 1), which provides insights into the Southern Hemisphere climate dynamics associated with DO-15.1 (see below). The construction of the age models is described in the Methods section. The comparison between the $\delta^{18}O_c$ records from the same and different caves in the same climatic realm (Figs. 1a–c and 2d–f) suggests that the EASM and ISM $\delta^{18}O_c$ records are broadly replicable despite some minor differences. The $\delta^{18}O_c$ record of speleothem SN23-1 from the Shennong cave is characterized by ~10-year resolution precisely constrained by U-Th dating (Fig. 1c), which is used to improve the accuracy of the bipolar ice-core chronologies.

It has been generally assumed that the $\delta^{18}O_c$ variability on annual- to orbital- timescales in the EASM and ISM domains primarily reflects changes in the large-scale monsoonal circulation and north-south shifts of the tropical rain belts. These processes are driven by upstream convection, moisture source dynamics, and overall monsoon intensity/circulation, rather than local rainfall amount alone[17,21,36–43] (Supplementary Note 1.1). Higher $\delta^{18}O_c$ values indicate a weaker EASM or ISM, and vice versa. At centennial to orbital timescales, lower $\delta^{18}O_c$ values are generally considered to reflect enhanced SASM intensity (overall monsoon circulation) and associated rainfall[15,44], and vice versa (Supplementary Note 1.1).

### Global synchronization and teleconnections

In order to compare our speleothem records to bipolar ice-core records on a common chronology, we tuned the Greenland ice-core records to our U-Th-based chronology using the correlation between the ASM $\delta^{18}O_c$ record and the Greenland ice-core [$Ca^{2+}$] record[22] (Supplementary Fig. 3a–c and Supplementary Table 1) (see "Methods"). This tuning suggests that the Greenland Ice-Core Chronology 2005 (GICC05) (Supplementary Note 1.2) requires a + 230-year shift around 55–56 ka BP, well within the reported ice-core age error estimate (~ 2350 years) in this time interval, to optimally align with the U-Th-based timescale ("Methods"). Moreover, by using the previously established match points between the North Greenland Ice-core Project (NGRIP) $\delta^{18}O$ and the West Antarctic Ice Sheet Divide ice core (WDC) $CH_4$ (Supplementary Fig. 3e and Supplementary Table 2), we correlated the WDC $CH_4$ record to the modified Greenland chronology (GICC05 + 230 years) during the studied time interval (Supplementary Note 1.4). A multi-proxy comparison based on a synchronized chronology shows that DO-15.1 as recorded by speleothem SN23-1 began at $55,191^{+90}_{-76}$ a BP, peaked at $55,141^{+75}_{-98}$ a BP, and then ended at $55,074^{+93}_{-83}$ a BP (Supplementary Fig. 4a), with a duration of ca. 120 years. This duration is similar to the duration of GI-15.1 in the Greenland ice-core records, which is ca. 100 years[5] (Supplementary Fig. 4b, c). Note that the timing of the onset of the speleothem record was determined using a similar method to that used for Greenland ice-core records ("Methods").

Corrick et al.[9], based on available ASM $\delta^{18}O_c$ records (Supplementary Fig. 5c–f) and European $\delta^{18}O_c$ records (Supplementary Fig. 5g, h), observed an asynchronous relationship between ASM and the North Atlantic during DO-15.1. They argued that this could be a result of insufficient resolution of the available records or that DO-15.1 may have been unusual compared to millennial-scale DO events. Our high-precision and much better resolved ASM speleothem records suggest that the timing of DO-15.1 in the ASM domain was synchronous (within combined $2\sigma$ uncertainties) with the European/North Atlantic region (Supplementary Fig. 5j, k) ("Methods"). By comparing detailed dating information with our SN23-1 record, we find that the ASM records used by ref. 9 are characterized by slower growth rates, larger dating errors, and lower dating resolution (Supplementary Table 3), which are likely the reason for the discrepancy in the timing of this DO event. Our results document the synchroneity of the onset of DO-15.1 between ASM and the North Atlantic realm, which underscores the fast atmospheric teleconnection, as shown for longer, i.e., millennial-scale, DO events[9,16,22].

The marine sediment Pa/Th ratio from the Bermuda Rise in the northwestern Atlantic Ocean (Supplementary Fig. 1), indicative of changes in the AMOC strength in this region, does not capture any fluctuations during DO-15.1, which is not surprising due to its low resolution (~ 160 years) between 57 and 54 ka BP (Fig. 2a). However, an interhemispheric comparison between marine and terrestrial records reveals that the climate variability within DO-15.1 was similar to that of longer DO events (Fig. 2b–g and Supplementary Fig. 4d–j), demonstrating the role of the AMOC also for short DO events. In particular, the SASM (Fig. 2g) and ASM $\delta^{18}O_c$ records (Fig. 2c–f) manifest an anti-phase relationship, indicating a weakening of the SASM and an intensification of the ASM during DO-15.1. This "monsoon seesaw pattern" is also coherent with a northward shift of the Intertropical Convergence Zone (ITCZ)[9,15,16,22], inferred from the Cariaco Basin sediment reflectance record (Supplementary Fig. 4h)[45]. These lines of evidence are consistent with large-scale atmospheric responses to changes in AMOC strength as shown by other proxy records and numerical modeling experiments[16,46–48].

In this study, we categorized DO events in MIS 3 into three distinct groups according to their durations in Greenland ice-core $\delta^{18}O$ records: (1) short DO events, exemplified by DO-15.1, which are characterized by a short duration (ca. 100 to 120 years) (Supplementary Fig. 4a–c); (2) intermediate-long DO events spanning several centuries up to 1200 years, and (3) super-long DO events lasting more than 1500 years, which typically occurred following Heinrich Stadials (Fig. 3). In the following discussion, both intermediate-long and super-long DO events are collectively considered as long DO events, in contrast to short DO events. Our classification of long and short DO events is also consistent with their respective underlying mechanisms: the short interstadials and the initial overshoot phases of long interstadials represent transient warming periods characterized by rapid, high-amplitude temperature increases in the northern high latitudes (Fig. 3a, b). These abrupt warmings reflect a transient phase when excess subsurface ocean heat accumulated during stadials is released (e.g., refs. 49,50), followed by a rapid cooling transition once this heat reservoir is depleted. In contrast, intermediate-length and super-long interstadials exhibit both this initial overshoot phase and a relatively stable active AMOC phase, where persistent northward oceanic heat transport sustained a relatively stable warm condition without additional subsurface heat release. Notably, the super-long interstadials that typically follow Heinrich Stadials may result from particularly large accumulations of subsurface heat during these severely cold periods[51], explaining their long durations.

At the onset of GI-15.1, the increase in Greenland ice-core $\delta^{18}O$ by ~6‰ (Fig. 2b), the decrease of ice-core [$Ca^{2+}$] by ~ 200 ppb (Supplementary Fig. 3b, c), and the decrease in ISM $\delta^{18}O_c$ by ~ 2.7‰ (Fig. 2e, f and Table 1) are comparable to respective changes at the beginning of

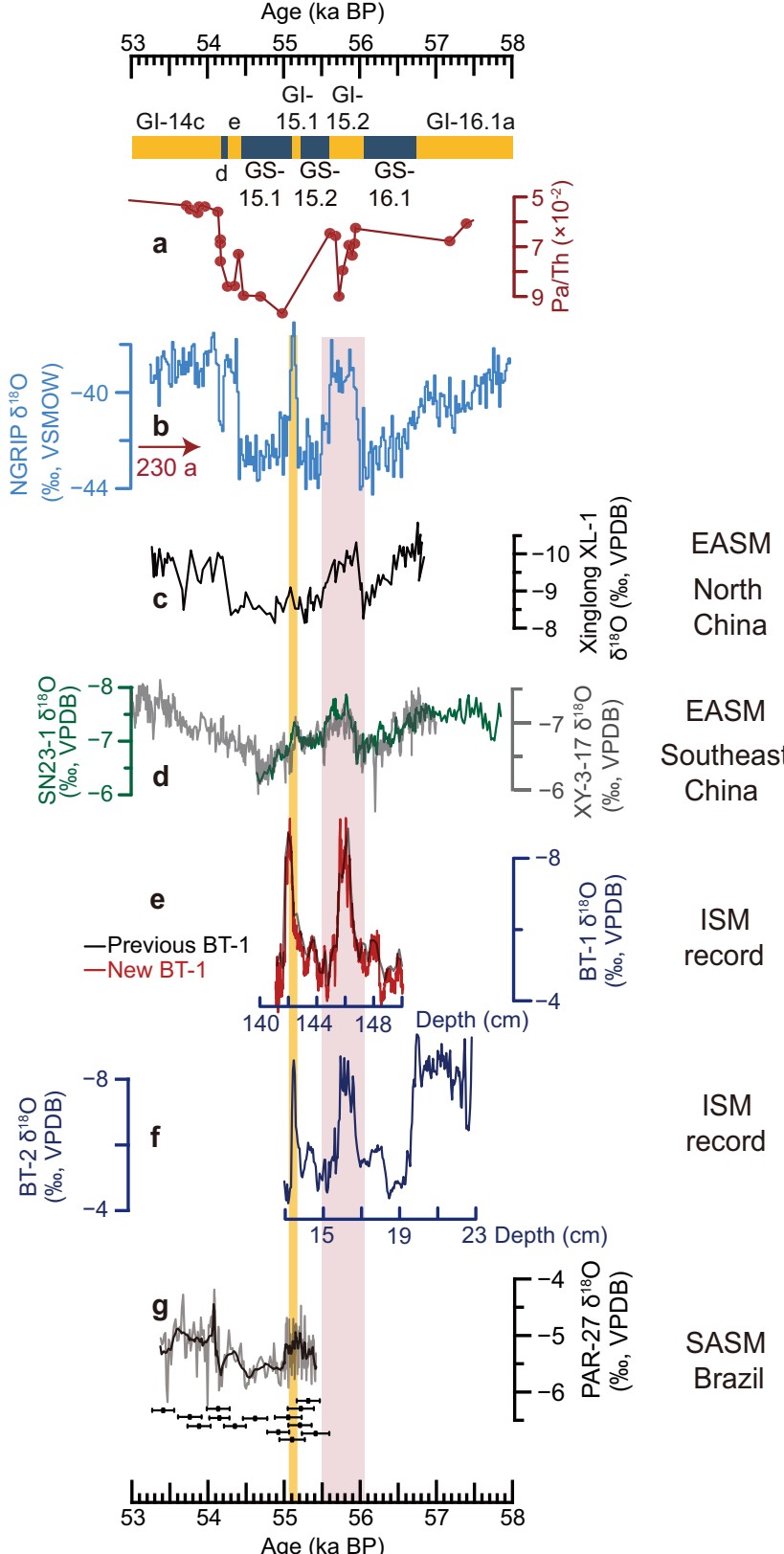

longer DO events (Fig. 3; Supplementary Fig. 3b, c). Therefore, our comparison of proxy records highlights the uniqueness of $\delta^{18}O_c$ over Southeast China with a smaller $\delta^{18}O$ depletion during DO-15.1 compared with long DO events (Figs. 2d and 3c and Table 1). This scenario is confirmed by another $\delta^{18}O_c$ record from Xianyun cave in Southeast China (Figs. 2d and 3c and Table 1)[52], suggesting that the hydroclimate

variability over Southeast China is likely related to the duration of DO events.

### EASM interstadial diversity linked to boreal Westerlies

To investigate the mechanisms responsible for the different interstadial magnitudes of $\delta^{18}O_c$ changes in Southeast China during DO

**Fig. 2 | Comparison between Greenland ice-core, marine sediment and speleothem records from different monsoon domains. a** $^{231}$Pa/$^{230}$Th from the Bermuda Rise (CDH19[11]), reflecting the strength of AMOC[11,98]. **b** Greenland NGRIP ice-core δ$^{18}$O records[4,5] on the GICC05 chronology and shifted by + 230 years. **c, d** Speleothem δ$^{18}$O record from Xinglong cave[99], Shennong cave (dark green, this study) and Xianyun cave[52] (gray) in the EASM domain. **e, f** Speleothem δ$^{18}$O records from Bittoo cave (navy, ref. 68; Red, this study) in the ISM domain. Note that the age uncertainty for the Bittoo cave records is large at this time (Supplementary

Table 3), thus this record is plotted against depth. **g** SASM speleothem record from Paraíso cave, Brazil (gray, original data; black, 10-point moving average; this study). Error bars show U-Th dates with uncertainties (2σ) for the record. Note the reversed y-axis in (**a**) and (**c–f**). Greenland Stadials (GS) and Greenland Interstadials (GI)[5] are shown at the top. The vertical bars indicate DO-15.1 and DO-15.2, respectively. Cave locations are shown Fig. 1d and Supplementary Fig. 1. *ISM* Indian summer monsoon; *EASM* East Asian summer monsoon; *SASM* South American summer monsoon. *DO* Dansgaard-Oeschger.

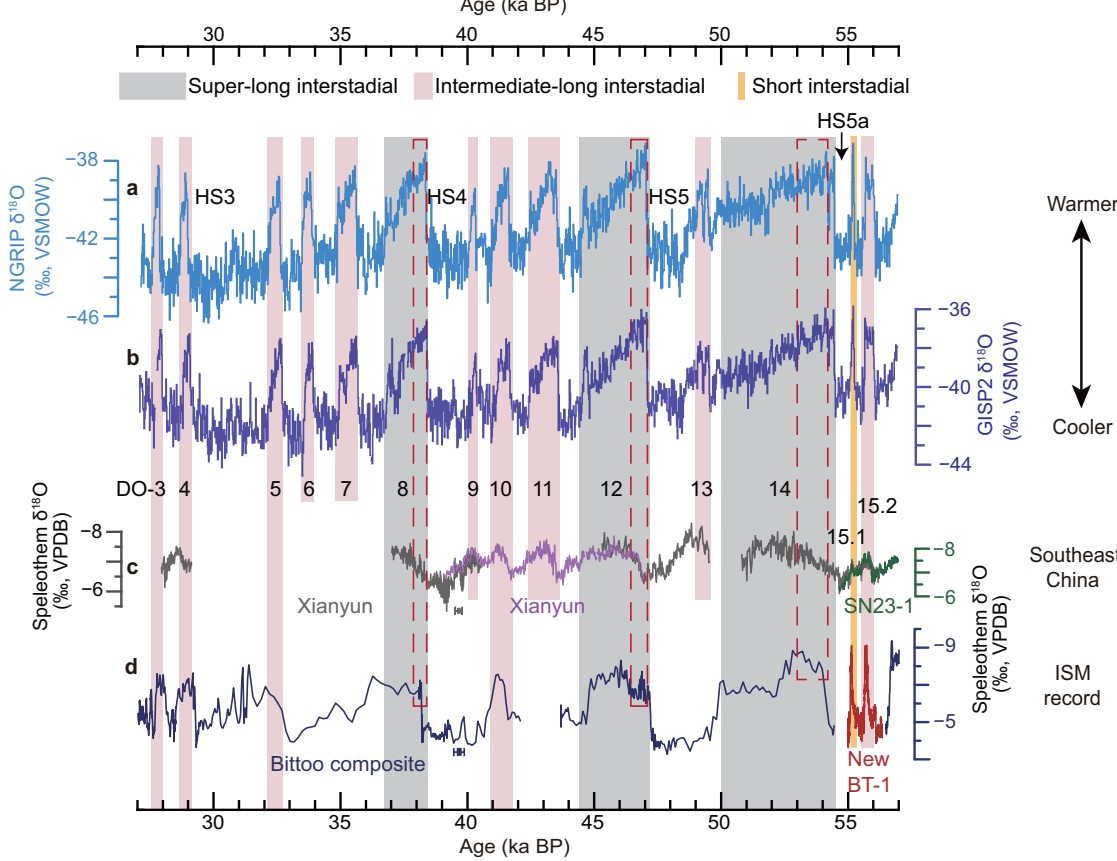

**Fig. 3 | Comparison between Greenland ice-core and speleothem records from ISM and EASM domains during Marine Isotope Stage 3. a, b** Greenland NGRIP[4,5] and GISP2[100,101] ice-core δ$^{18}$O records, respectively, plotted on the 1.0063 × GICC05 chronology[102] (Supplementary Note 1.4). **c** Southeast China speleothem δ$^{18}$O records from Shennong cave (this study) and Xianyun cave[52,103–106], respectively. Note that the XY-35 time series (purple) from Xianyun cave has been shifted by + 0.3‰ in order to be in agreement with the other records from this cave. **d** ISM domain Bittoo cave speleothem δ$^{18}$O record [navy, ref. 68; Red, this study, plotted on a depth scale as in Fig. 2e]. Error bars in (**c**) and (**d**) indicate the typical age model

uncertainties (2σ) of the Xianyun and Bittoo records, respectively. The vertical bars indicate short DO (DO-15.1), intermediate-length DO (lasting a few hundred years) and super-long DO (duration more than 1500 years), respectively (see main text). Note that we adopted the labeling convention DO-X for interstadials identified in both speleothem and ice-core records, where X corresponds to the numbering of Greenland Interstadials (GI) as outlined by Rasmussen et al.[5]. The red dashed boxes show the "overshoot" phases of the super-long interstadials. *HS* Heinrich Stadial; *ISM* Indian summer monsoon; *EASM* East Asian summer monsoon. *DO* Dansgaard-Oeschger.

events of different durations, we conducted an AMOC self-oscillatory experiment against the climate background of MIS 3[53] (Fig. 4). We used the isotope-enabled COSMOS-wiso climate model, whose performance for abrupt climate changes during glacial periods[53–56] and water isotopic variations in the EASM domain[43,57] have been well studied ("Methods"). In our experiment, the simulated AMOC is characterized by a brief overshoot phase lasting several decades at the onset of an interstadial, followed by a relatively stable and strong interstadial mode before the abrupt transition to a weak stadial mode (Fig. 4a). This characteristic "sawtooth" structure[2,3,5,58] is well documented in Greenland ice-core δ$^{18}$O records (NGRIP and GISP2) from DO-3 to DO-14 (Fig. 3a, b), and matches the observed behavior of most long interstadials. We therefore use the AMOC overshoot and the interstadial phase as analogues for short and long DO events, respectively, in our experiments (Fig. 4a; "Methods"). In our simulations, the long

interstadials were specifically defined to incorporate the overshoot phases (Fig. 4a), consistent with proxy records (Fig. 3a, b). This design enables us to effectively analyze how interstadial durations modulate responses of the hydroclimate variability in the EASM region during DO events.

The simulated ratio of precipitation δ$^{18}$O ($δ^{18}O_p$) changes, expressed as (Interstadial minus Overshoot phases)/(Interstadial minus Stadial phases), an indicator of the impact of the Northern Westerlies on interstadial ASM $δ^{18}O_p$ variations, shows a similar pattern to the $δ^{18}O_c$ records between long and short DO events (Fig. 4d and Table 1). Specifically, this indicates less depleted $δ^{18}O_p$ in Southeast China during the overshoot phase in comparison with the interstadial phase (Figs. 2d and 3c and Table 1), in contrast to the recorded small δ$^{18}$O differences in northern India (Figs. 2e, f and 3d and Table 1). In contrast to the tropical ISM system, the subtropical EASM system is

**Table 1 | The event amplitude of δ¹⁸O changes in different cave records**

| Cave | Event-type | DO event | $\delta^{18}O$ amplitude[a] | Ratio |
|---|---|---|---|---|
| Southeast China | | | | |
| Shennong | Short DO | DO-15.1 | 0.2 | $\frac{\delta^{18}O_{(INTER-Short)}}{\delta^{18}O_{(INTER)}} = 0.67$ |
| | INTER DO | DO-15.2 | 0.6 | |
| Xianyun | INTER DO | DO-4 | 0.4 | $\frac{\delta^{18}O_{(SUPER-INTER)}}{\delta^{18}O_{(SUPER)}} = 0.47$ |
| | | DO-9 | 0.3 | |
| | | DO-10 | 0.6 | |
| | | DO-11 | 0.6 | |
| | | DO-15.2 | 0.4 | |
| | SUPER DO | DO-8 | 0.8 | |
| | | DO-12 | 0.9 | |
| | | DO-14 | 0.9 | |
| ISM domain | | | | |
| Bittoo | Short DO | DO-15.1 | 2.7 | $\frac{\delta^{18}O_{(INTER-Short)}}{\delta^{18}O_{(INTER)}} = -0.27$ |
| | INTER DO | DO-3 | 1.8 | |
| | | DO-4 | 2 | |
| | | DO-15.2 | 2.6 | |
| | SUPER DO | DO-8 | 2.2 | $\frac{\delta^{18}O_{(SUPER-Short)}}{\delta^{18}O_{(SUPER)}} = -0.06$ |
| | | DO-12 | 2.9 | |

[a] The amplitude of δ¹⁸O changes were calculated based on the Mean-fitting algorithm ("Methods"; Supplementary Fig. 9).
*DO* Dansgaard-Oeschger. *INTER DO* Intermediate-long DO; *SUPER DO* Super-long DO. *ISM* Indian summer monsoon.
Regarding the ratio, we have rounded it to two decimal places.

additionally modulated by the location of the summer mid-latitude Westerlies ("Methods"), which itself is controlled by the meridional thermal gradient[28,29,59]. In both the overshoot and interstadial phases, the strengthened AMOC warms up the northern high latitudes, reducing the meridional thermal gradient and hence promoting a northward shift of the Westerlies (Fig. 5c and Supplementary Fig. 6i). However, during the overshoot phase, the release of additional heat in the subsurface North Atlantic, which accumulated during cold stadial periods, leads to additional high-latitude warming (e.g., refs. 49,50) (Figs. 3a, b, 5d, 6b). This causes a further northward displacement of the Westerlies (Figs. 5d and 6b), promoting a northwestward expansion of the Western Pacific subtropic high (Fig. 5a), transporting more water vapor from the western Pacific to the EASM region (Figs. 4b, c and 6b). As a result, more near-source water vapor from the western Pacific with enriched oxygen isotopic composition would effectively reduce the magnitude of the isotopic depletion in EASM precipitation during short DO warmings (e.g., DO-15.1) (Fig. 4b, c), thus accounting for the contrasting speleothem δ¹⁸O signals between DO-15.1 and long DO events.

Our simulation results, which demonstrate the shift of the Westerlies (Fig. 5c and Supplementary Fig. 6i), exhibit broad consistency with multiple lines of evidence. These include existing paleoclimate reconstructions derived from marine, speleothem, and lacustrine archives within Westerlies-dominated regions, as well as outputs from various climate model simulations ("Methods"). Previous reconstructions have consistently indicated a systematic meridional displacement of the boreal Westerlies, characterized by southward shifts during stadials and northward migrations during interstadials ("Methods"). Building upon these established patterns, our proposed mechanism provides a more comprehensive

framework that elucidates not only the distinction between long and short interstadials but also the temporal evolution within long interstadials, particularly between the initial overshoot phase and subsequent periods. Within this dynamic framework, we predict less depleted δ¹⁸O values in Southeast China during the overshoot phase of long interstadials, indicative of a more pronounced northward shift of the Westerlies. This prediction is strongly supported by the Xianyun cave record from Southeast China, which reveals a distinct isotopic pattern: the δ¹⁸O values during the overshoot phases of DO-8, DO-12, and DO-14 are systematically less depleted compared to those recorded in their respective subsequent periods (Fig. 3c). In contrast, the Bittoo cave $\delta^{18}O_c$ record from the ISM domain shows minor differences between the overshoot and subsequent interstadial phases (Fig. 3d). This empirical evidence corroborates our model predictions and strengthens the proposed mechanism during these climatic transitions— that is, the intensity of high-latitude warming, rather than the duration of interstadials alone, is the essential factor driving the northward shift of the Northern Westerlies during interstadials.

It is noteworthy that the $\delta^{18}O_c$ manifests spatial heterogeneity in the vast ASM domain. Yongxing and Xiaotian cave $\delta^{18}O_c$ records show small differences between long and short DO events (Fig. 1b, d and Supplementary Fig. 5c, d). Similarly, the Qixing and Wulu cave $\delta^{18}O_c$ records, located in the transitional zone between the EASM and ISM regions (Fig. 1d), also exhibit minor differences between long and short DO events (Fig. 1a and Supplementary Fig. 5e), despite of relatively low resolutions of the records that prevent further statistical analyses. In contrast, the Xinglong cave $\delta^{18}O_c$ record from northern North China (Fig. 1d) shows a significant difference between short and long DO events, similar to records from Southeast China (Fig. 2c). This spatial heterogeneity is broadly consistent with our model simulations that reproduce fairly similar spatial patterns (Fig. 4d). Of note is that summer rainfall in North China decreases northward (Supplementary Figs. 5i, 6d, e), which increases the sensitivity of $\delta^{18}O_p$ to rainfall variations northward[17,60]. As a result, even minor changes in rainfall amount and moisture sources can lead to substantial variations in $\delta^{18}O_p$ in North China, thus causing a complex pattern in this region (Fig. 4d). Taken together, our analyses highlight a broad agreement between model simulations and proxy records, providing a coherent framework for the diverse responses of the Westerlies-ASM dynamics to short and long DO events.

In order to further investigate whether the presence of the "overshoot" phase affects our conclusions, an alternative "interstadial" phase by excluding the prior "overshoot" phase was defined (Supplementary Fig. 7a; "Methods"). These alternative results align well with the simulated results that include the overshoot phase (Figs. 4 and 5 and Supplementary Fig. 7), consistently revealing minimal differences between the long and short interstadials over India and significant differences over Southeast China (Fig. 4d and Supplementary Fig. 7c), and therefore further strengthening our conclusions.

In conclusion, our findings highlight the uniqueness of the proxy footprint ($\delta^{18}O_p$) of EASM hydroclimate, which is modulated by mid-latitude processes in addition to tropical processes, distinguishing it from $\delta^{18}O_p$ changes over the tropical ISM system. Specifically, based on empirical and theoretical investigations of short (DO-15.1) and long DO events in MIS 3, we provide a coherent dynamic framework that explains the different responses of $\delta^{18}O_p$ in the ASM region to the short (or overshoot phases) and long interstadials. Moreover, our study suggests that the GICC05 chronology requires a +230-year shift between 55–56 ka BP to align with the U-Th-based speleothem chronology. Comparison of proxy records on the synchronized chronology suggests that the AMOC is likely the driving factor for both short and long DO events.

For the future, it is highly desirable to reconstruct more high-resolution records across eastern China to confirm the proposed interstadial diversity of EASM hydroclimate and spatially nuanced responses of EASM $\delta^{18}O_p$ changes across different DO events. Our

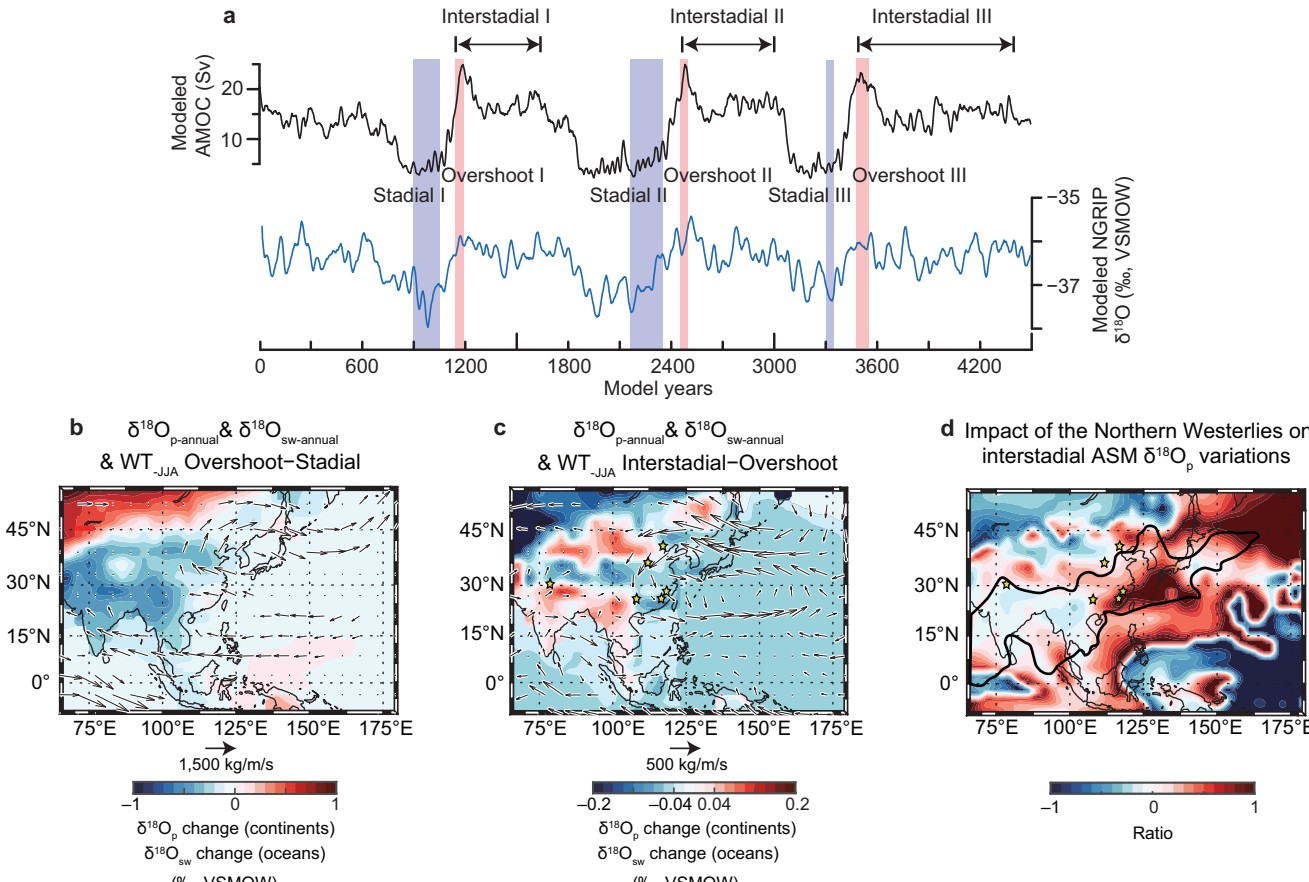

Fig. 4 | Simulated hydroclimate changes during DO warmings. a Modeled AMOC index defined by the maximum stream function between 500–2500 m in the North Atlantic north of 45°N in the self-oscillatory AMOC equilibrium experiment (E40ka_34Orb in ref. 53). We classify three AMOC phases: Stadial phases, Overshoot phases and Interstadial phases (Methods). Also shown is the modeled NGRIP mean annual $\delta^{18}O_p$. Both time series were averaged using a 30-year running window. b Composites of changes in mean annual $\delta^{18}O_p$ over continents and mean annual $\delta^{18}O$ of sea-surface water ($\delta^{18}O_{sw}$) over oceans (shaded, units: ‰) and vertical integrated water-vapor transport (WT) (vectors, units: kg/m/s) during simulated short (Overshoot minus Stadial phases) warming events, (c) same as (b) but for differences between simulated long and short (Interstadial minus Overshoot phases) interstadials. Note that the color scales and wind vectors in (b) and (c) are different. d Impact of the Northern Westerlies on interstadial ASM $\delta^{18}O_p$ variations,

expressed as the ratio of $\delta^{18}O_p$ changes: (Interstadial minus Overshoot phases)/(Interstadial minus Stadial phases). In the core ASM domain, positive values denote regions with influences of the Westerlies shifts. The values of the ratio approaching 1 indicate a muted $\delta^{18}O_p$ depletion during transitions from stadial to short interstadials and to overshoot phases of long interstadials, as a consequence of a further northward shift in Northern Westerlies. Ratios near zero indicate $\delta^{18}O_p$ differences between interstadials and stadials are independent of the Westerlies shift. Bittoo, Xinglong, Xiaotian, Qixing, Shennong and Xianyun cave locations are indicated by the yellow stars (refer to Fig. 1d). The thick black line delineates the approximate region of the EASM and ISM and the transitional zone between them, as same as in Fig. 1d. ASM Asian summer monsoon; ISM Indian summer monsoon; EASM East Asian summer monsoon. DO Dansgaard-Oeschger.

simulations reveal that the degree of high-latitude warming – rather than interstadial duration alone – modulates the expression of DO variability in some regions. Our findings provide clear evidence that the Westerlies' position is not simply binary (north during all interstadials versus south during stadials), but rather exhibits a continuum of responses depending on the degree of high-latitude warming. This continuum reflects the underlying ocean-atmosphere dynamics – from the intense but short-lived heat release during overshoots to the more sustained but moderate forcing during quasi-equilibrium phases. This highlights the need for future studies to take into account the influence of high- and low-latitude meridional temperature gradients on the East Asian hydroclimate. Lastly, it can be concluded that although the overall response of the global monsoon system to DO events is consistent across large regions, there are subtle differences among different monsoon systems. Therefore, it is necessary to analyze the individual climate characteristics of each DO event when studying some oceanic and continental climate systems, in addition to analyzing the overall characteristics of large regions and multiple events.

## Methods

### ISM and EASM climate dynamics

The different heating of the atmospheric column between continents and oceans drives planetary-scale circulation patterns[33]. As for the ASM, its onset is closely accompanied by a northward shift of the ITCZ on the seasonal scale[20,47]. The ASM transports moisture and heat from tropical oceans northward to South Asia and East Asia (Fig. 1d and Supplementary Fig. 5i), bringing abundant summer monsoon rainfall. Notably, the ASM shows both dynamic and thermodynamic aspects, and in this study, we mainly focus on the dynamics of the ASM circulation.

The vast ASM system consists of three major subsystems, the ISM, the EASM and the Western North Pacific summer monsoon (WNPSM) systems[18,19] (Fig. 1d), and this study focuses on the former two subsystems. The ISM is essentially a tropical system[19,33,61], characterized by a strong low-level southwesterly jet, which extends from the Mascarene High in the Southern Hemisphere across the Arabian Sea and the Bay of Bengal to the north and northeast of India (Fig. 1d). The deep convections in the system start in mid-May, followed by the onset of

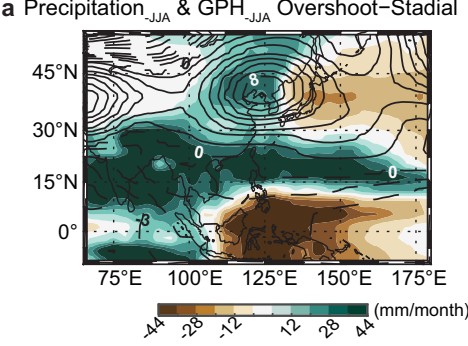

**a** Precipitation$_{\text{-JJA}}$ & GPH$_{\text{-JJA}}$ Overshoot−Stadial

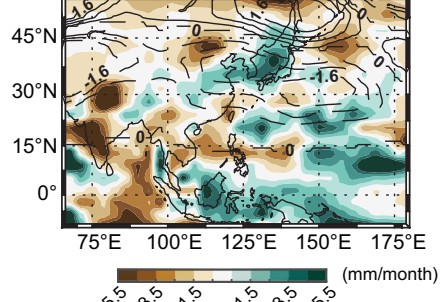

**b** Precipitation$_{\text{-JJA}}$ & GPH$_{\text{-JJA}}$ Interstadial−Overshoot

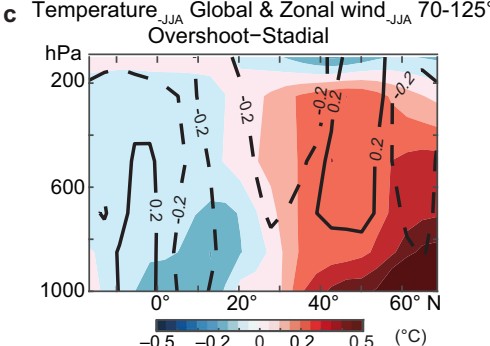

**c** Temperature$_{\text{-JJA}}$ Global & Zonal wind$_{\text{-JJA}}$ 70-125°E Overshoot−Stadial

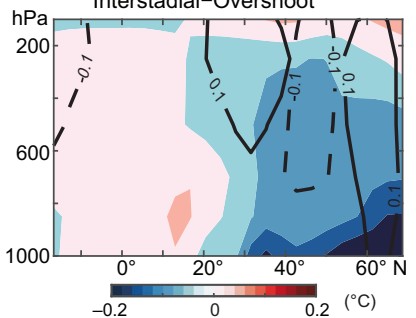

**d** Temperature$_{\text{-JJA}}$ Global & Zonal wind$_{\text{-JJA}}$ 70-125°E Interstadial−Overshoot

**Fig. 5 | Governing mechanisms of EASM interstadial diversity. a** simulated change in precipitation (shading, units: mm/month) and 500 hPa geopotential height (lines) during short DO warming events (Overshoot minus Stadial phases), (**b**) same as (**a**) but for the differences between long and short interstadials (Interstadial minus Overshoot phases). The intervals between each line correspond to 1 and $\frac{8}{15}$ in (**a**) and (**b**), respectively. **c**, **d** are as same as (**a**) and (**b**), respectively, but for global zonal-averaged summer temperature (shadings) and zonal-averaged wind between 70° and 125° E (lines). The time periods of simulated Stadial, Overshoot and Interstadial phases refer to Fig. 4a. Note that the solid lines indicate positive values while dashed lines indicate negative values. *EASM* East Asian summer monsoon. *DO* Dansgaard-Oeschger.

the ISM in June. The mid-latitude westerly jet stream is generally not of first-order importance[33]. Indeed, the classic Gill model, which takes into account the displacement of heat sources around north of the equator to mimic the summer monsoon circulation[62], captures the major features of this tropical monsoon system.

The EASM is effectively a subtropical monsoon system, characterized by low-level southwesterly and southerly moisture and heat transport over the low- and mid-latitudes of East Asia (Fig. 1d). The westerly jet is crucial for the EASM, while the TP is an important orographic forcing that steers the westerly jet stream[28–31]. During spring, the jet position swings between the north and south of the TP. During early summer, a decrease in the temperature gradient between high- and low-latitudes pushes the jet stream to the north of the TP, triggering a northward jump of the EASM. This stage is characterized by the Meiyu-Baiu rain belt extending from the mid- and lower reaches of the Yangtze River to Japan[63]. During late July, the westerly jet stream jumps further northward, and the Meiyu-Baiu rain belt vanishes, allowing the southerly-brought moisture to be transported further northward to the deep interior of the continent[29,31].

## Cave settings and modern climatology
This study is based on five δ$^{18}$O$_c$ records, one from the SASM domain and four from the ASM domain. Speleothem PAR-27 was obtained from Paraíso cave (4.07°S, 55.45°W; 60 m a.s.l.) (Fig. 2g)[15,44] in the Amazon Basin, SASM domain (Supplementary Fig. 1); QX-4 from Qixing cave (25.98°N, 107.26°E, 1,020 m a.s.l.) with isotope data reported in ref. 64 and U-Th dates provided by this study (Fig. 1a); XT-5 came from Xiaotian cave (35.47°N, 112.03°E, 1,460 m a.s.l.)[65] (Fig. 1b); SN23-1 from

Shennong cave (28.7°N, 117.25°E, 380 m a.s.l.)[43,66,67] (Fig. 1c); and BT-1 from Bittoo cave (30.78°N, 77.03°E, ~3000 m a.s.l.) with U-Th dates reported in ref. 68 and isotope data provided by this study (Fig. 2e). Cave locations in the ASM domain are shown in Fig. 1d.

The mean annual air temperature at the meteorological station near Xiaotian cave is 12.6 °C, and the mean annual rainfall in this region is ~ 480 mm[65]. The mean annual air temperature at the meteorological station near Qixing cave is 16.6 °C, and the mean annual rainfall in this region is ~ 1180 mm[64]. At these two cave sites, more than 60% of the annual rain falls during the summer months (June to September)[64,65]. The mean annual air temperature at the meteorological station near Shennong cave is ca. 19 °C, and the mean annual rainfall in this region is ~ 2050 mm[66,67]. The summer monsoon season contributes nearly one-half of annual rainfall, while the spring persistent rainfall (March to May) is also an important contributor in this region[66,67,69]. The mean annual rainfall at the Bittoo cave site is ~ 1600 mm, with more than 80% of the annual rain falls during the summer months (June to September)[68].

The mean annual temperature near Paraíso cave is about 26 °C, and the annual precipitation exceeds 2000 mm, with nearly 70% falling during the SASM season (the wet season, November to April)[44].

## U-Th dating
Subsamples for U-Th dating were drilled on the polished speleothem section using a 0.3-mm carbide dental drill. We used standard chemistry procedures[70] to separate U and Th. A triple-spike ($^{229}$Th-$^{233}$U-$^{236}$U) isotope dilution method was used to correct instrumental fractionation and to determine U-Th isotopic ratios and concentrations[71,72].

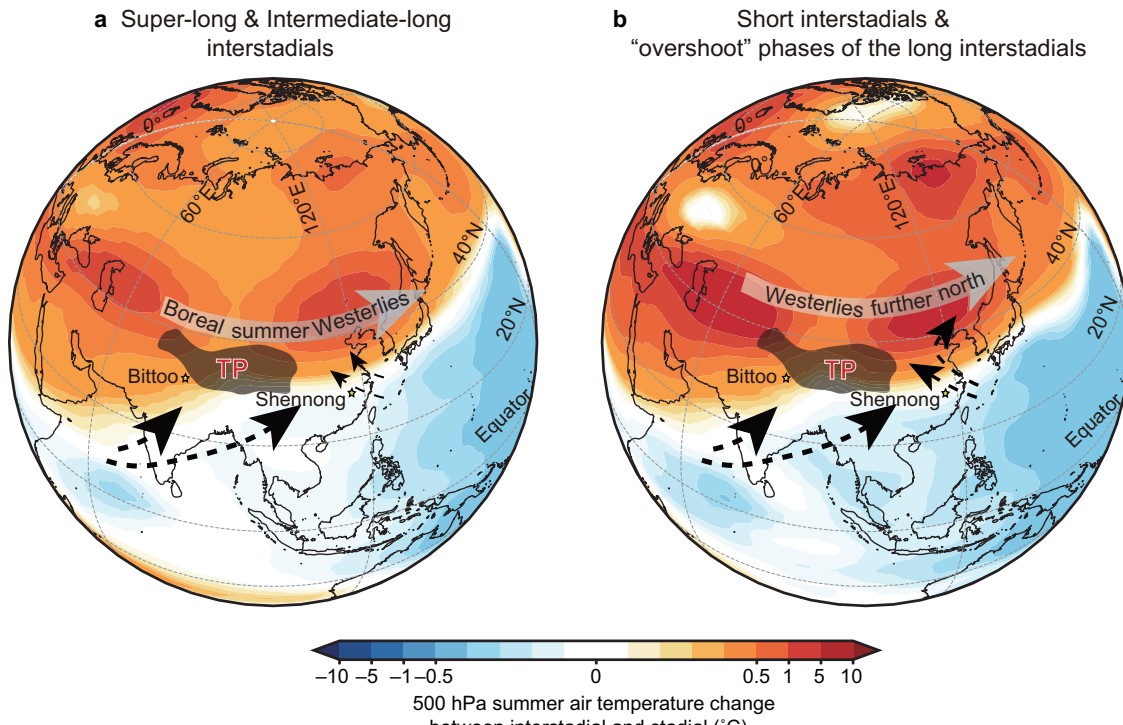

**a** Super-long & Intermediate-long interstadials

**b** Short interstadials & "overshoot" phases of the long interstadials

500 hPa summer air temperature change between interstadial and stadial (°C)

**Fig. 6 | Schematic diagram of the boreal Westerlies-modulated nuances of East Asian water vapor transport during short and long interstadials. a** Summer air temperature change at 500 hPa (shading) between long interstadial and stadial (simulated Interstadial minus Stadial phases), and schematic boreal Westerlies (gray arrow) and low-level water vapor transport (black arrows) during the long DO warming events. **b** is the same as in (**a**), but the temperature change is between short interstadial and stadial (simulated Overshoot minus Stadial phases), and Westerlies and low-level water vapor transport are during the short DO warming events. The time periods of simulated Overshoot, Interstadial and Stadial phases refer to Fig. 4a. The definition of short, intermediate-long and super-long interstadials refers to Fig. 3. TP indicates the approximate location of the Tibetan Plateau. *DO* Dansgaard-Oeschger.

$^{230}$Th dating was performed at Xi'an Jiaotong University, China, using a Thermo-Finnigan Neptune Plus multi-collector inductively coupled plasma mass spectrometer (MC-ICP-MS). U and Th isotopes were measured on a MasCom multiplier behind the retarding potential quadrupole in peak-jumping mode using standard procedures[71]. Uncertainties in U and Th isotopic measurements were calculated offline at the 2σ level, including corrections for blanks, multiplier dark noise, abundance sensitivity and contents of the same nuclides in the spike solution. The most recent values for the decay constants of $^{234}$U and $^{230}$Th[72] and $^{238}$U[73] were used. Corrected U-Th ages assume an initial $^{230}$Th/$^{232}$Th atomic ratio of $(4.4 \pm 2.2) \times 10^{-6}$, and those are the values for a material in secular equilibrium with the bulk earth $^{232}$Th/$^{238}$U value of 3.8. Because the samples used in this study have high U concentrations (~ 200–35,000 ng/g) and the influence of detrital $^{232}$Th is small (Supplementary Data 1), the corrected ages are not particularly sensitive to the initial Th isotope ratio.

　　We obtained a total of 46 U-Th dates, 19 from SN23-1, 14 from PAR-27, 10 from QX-4 and 3 from XT-5 (Supplementary Data 1). We also used two published U-Th dates from QX-4[64] and XT-5[65] (Supplementary Data 1). Typical age uncertainties (2σ) vary between 90 and 180 years for the key intervals (Supplementary Data 1).

**Oxygen isotope analysis**
A total of ~1,650 subsamples were drilled for stable isotope analyses (δ$^{18}$O). 354 from SN23-1 (micro-milled at a spatial resolution varying between 0.25 and 1 mm), 252 from PAR-27 at a spatial resolution of 1 mm, 163 from XT-5 (micro-milled at a spatial resolution of 0.2 mm) and ~ 880 from BT-1 (micro-milled at a spatial increment of 0.1 mm). Analyses were carried out at Xi'an Jiaotong University using a Thermo-Finnigan MAT-253 mass spectrometer fitted with a Kiel IV Carbonate Device. δ$^{18}$O values are reported in per mil (‰), relative to the Vienna

Pee Dee Belemnite (VPDB) standard, δ$^{18}$O = [($^{18}$O/$^{16}$O)$_{sample}$/($^{18}$O/$^{16}$O)$_{VPDB}$ − 1] × 1000‰. The analytical precision of the δ$^{18}$O analyses in this study is typically better than 0.1‰ (1σ).

**Age models**
To establish age models for speleothem records, we employed the StalAge algorithm[74] (Supplementary Fig. 2). StalAge is a suitable algorithm known for creating objective age models based on two assumptions: (1) the age model is monotonic, and (2) a straight line is fitted through as many data points as possible within error bars[74]. Through Monte-Carlo simulations, StalAge generates 300 realizations of age models to account for the 95% confidence limits[74]. Major outliers are detected by disagreement with at least two data points, while minor outliers are screened if more than 80% of the simulated straight lines fail to have a positive slope. In our study, we did not identify any major or minor outliers, as all ages in each age model increase monotonically within the dating uncertainties (Supplementary Fig. 2). We also calculated age models by using other age modeling schemes that yielded virtually identical results, and thus the speleothem age models established here are insensitive to the choice of the algorithms (Supplementary Fig. 2).

**Identification of the onset of DO-15.1**
In this study, we applied the method used to identify the onset of Greenland interstadials[5] to determine the onset timing of DO-15.1 in the SN23-1 record. Therefore, the onset of DO-15.1 is taken to be "*the first data point of the steep part that clearly deviates from the baseline level preceding the transition*[5]". The baseline level preceding DO-15.1 is shown by the horizontal dashed line in Supplementary Fig. 4a, thus, the age of the onset of DO-15.1 is defined as the initial data point that exceeds the baseline level (Supplementary Fig. 4a).

                    

## Change point determination and sensitivity tests

To objectively identify change points in various records, the BREAKFIT algorithm was utilized[75] (Supplementary Table 1). BREAKFIT identifies a single change point and fits a linear slope on either side, making it suited for determining the change points in SN23-1 $\delta^{18}O$ and Greenland ice-core [Ca$^{2+}$] records. The algorithm utilizes 2000 block bootstrap simulations to provide statistical uncertainties for the timing of breakpoints and considers the influence of autocorrelation for uneven time spacing[75]. We conducted sensitivity tests in which the search time windows were randomly changed (Supplementary Fig. 8). Results were deemed robust only if the timing of the identified change points did not vary by more than 30 years when the width of the search time window changed. The results indicate that the change points are insensitive to variations in the search time window.

## Phasing relations between climate events in the ASM domain and Europe

We analyzed the phasing of DO-15.1 between the North Atlantic region and the ASM domain using $\delta^{18}O_c$ records from Hölloch cave (Austria), Qixing and Shennong caves (China) (Fig. 1a, c; Supplementary Fig. 5g). Our analysis suggests that the timing of DO-15.1 recorded in the ASM domain is synchronous (within combined 2σ uncertainties) with the North Atlantic region (Supplementary Fig. 5j, k). A similar pattern has been observed in the previous studies for millennial-scale events[9]. We demonstrated the interregional synchrony of the DO-15.1 by comparing the onset timing (Supplementary Fig. 5j) and vertex points (Supplementary Fig. 5k). Therefore, these teleconnections appear to be robust regardless of their durations on millennial or centennial timescales.

It is noted that the onset timing of DO-15.1 from our ASM records is ca. 55.2 ka BP (Supplementary Fig. 5a), older than the result (54.74 ka BP) generated by ref. 9. This is because the timing of DO-15.1 in the previously used ASM records is too young (Supplementary Fig. 5d–f), which results in the discrepancy in the reconstructed timing of this DO event between different regions.

The widely used Hulu Cave $\delta^{18}O_c$ record appears to lack the distinct structure of DO-15.1 (Supplementary Fig. 5a). This absence could be attributed to several factors, including the relatively small amplitude of the event in the region (Fig. 4d), a slower growth rate or a hiatus during the particular time period, or the influence of epikarst processes. These possibilities highlight the need for further investigation to elucidate the underlying causes.

## Synchronizing the Greenland chronology to the U-Th chronology

The Greenland ice-core records over the past 60 ka rely largely upon the annual-layer counted GICC05 chronology (Supplementary Note 1.2). In spite of the fact that ice-core annual-layer counting provides precise relative ages, counting uncertainty accumulates with age, resulting in a maximum counting error of ± 2350 years around 55 ka BP. As a result of the large reported error estimate, it is difficult to correlate Greenland ice-core records adequately well with other paleoclimatic records.

A recent study[22] proposed a correlation strategy between the ASM domain $\delta^{18}O_c$ records and the Greenland ice-core [Ca$^{2+}$] record (a dust proxy[76]), which suggests that these two proxy records vary synchronously within decadal errors during the late last glacial period. This synchronicity can be attributed to the fact that the Taklimakan Desert and the Gobi Desert in Asia served as the primary dust sources to central Greenland during the last glacial period[77–79]. The dust transport via the Northern Hemisphere Westerlies is closely coupled with the large-scale ASM circulation[21,28]. As such, the correlation between Greenland ice core [Ca$^{2+}$] and ASM $\delta^{18}O_c$ records is causally plausible, which is further substantiated by the extant volcanic evidence and radiocarbon ages (ref. 22 and references therein), as well as by recent studies[80,81].

Therefore, based on the correlation strategy between the Greenland ice-core [Ca$^{2+}$] record and the SN23-1 $\delta^{18}O$ record, we were able to determine three tie points: (i) 55.14 ± 0.1; (ii) 55.66 ± 0.1 and (iii) 55.97 ± 0.11 ka BP (Supplementary Fig. 3; Supplementary Table 1). Our analysis suggests that the Greenland ice-core GICC05 chronology requires a + 230-year shift between 55 and 56 ka BP, well within the ice-core age uncertainty (~ 2350 years) (Supplementary Fig. 3g).

Our proposed age adjustment for GICC05 is also consistent within uncertainties with the previous speleothem tie point from ref. 82 and the radionuclide tie point from ref. 83, but with significantly improved precision and more tie points (Supplementary Fig. 3g). Overall, the speleothem-based tuning of the Greenland ice-core chronology reduces its absolute age uncertainty by an order of magnitude (Supplementary Fig. 3g and Supplementary Note 1.3).

## "Mean-fitting" algorithm

In this study, we employed an approach termed the "Mean-fitting" algorithm to determine the event amplitudes of $\delta^{18}O$ changes (Supplementary Code 1). This algorithm, developed in MATLAB, computes the average values for both the pre-transition and post-transition phases, represented by fitted horizontal lines, and simultaneously identifies the transitional change points. This method eliminates the need for predefined search time windows; however, the fitting results may be influenced by the specified maximum number of change points. The outcomes of the Mean-fitting analysis are illustrated in Supplementary Fig. 9, with the corresponding event amplitudes detailed in Table 1.

## Climate model description

**Model settings.** We used a fully coupled atmosphere–ocean general circulation model (AOGCM), COSMOS (ECHAM5-JSBACH-MPI-OM). The atmospheric model ECHAM5[84], complemented by the land surface component JSBACH[85], was used at T31 resolution (~ 3.75°), with 19 vertical layers. The ocean model MPI-OM[86], including sea-ice dynamics formulated using viscous-plastic rheology[87], has a resolution of GR30 (3° × 1.8°) in the horizontal, with 40 uneven vertical layers. Water isotopes have been implemented in COSMOS to enable a direct comparison between simulated and observed isotopic values[57]. In this study, we employed the equilibrium experiment (E40ka_34Orb) with ice volume and atmospheric $CO_2$ concentrations (195 ppm) fixed at an equilibrated control simulation (E40ka_CTL), while the orbital settings were set to 34 ka BP[53]. The specific orbital parameters are as follows: the eccentricity is 0.014996, the precession of the equinoxes (the angle between the Earth's position during the Northern Hemisphere vernal equinox and the orbit perihelion) is 84.84°, and the obliquity is 22.6°. The model was integrated for 5500 years to explore the equilibrated responses of the glacial climate to orbital configurations at 34 ka BP[53]. Notably, 34 ka BP is a time when internal climate background conditions (ice volume[88] and atmospheric $CO_2$ level[89]) were relatively constant and similar to that at 40 ka BP.

The E40ka_34Orb experiment has the characteristics of millennial-scale AMOC self-oscillation, resembling recorded DO cycles during MIS 3[53] (Figs. 3a, 4a). The COSMOS model has already been used to investigate a range of paleoclimate phenomena, especially millennial-scale abrupt glacial climate changes such as DO events[53–56,90], therefore it is a suitable model for this study.

**Definition of different phases.** In this study, the modeled AMOC index (Fig. 4a) is defined by the maximum stream function between 500–2500 m in the North Atlantic north of 45°N. We classified three AMOC phases: Stadial phases (900–1050, 2150–2350 and 3300–3350 model years), Overshoot phases (1150–1200, 2450–2500 and 3480–3550 model years) and Interstadial phases (1150–1650, 2450–3000 and 3480–4400 model years) (Fig. 4a). In our simulations, the long interstadials were specifically designed to incorporate the

overshoot phases, a feature consistent with observations of most long interstadials in Greenland (Fig. 3a, b).

To further investigate whether the presence of the "overshoot" phase might influence our conclusions, an alternative "interstadial" phase was defined by excluding the prior "overshoot" phase. In this scenario, the stadial and overshoot phases remain unchanged, while the interstadial phases changed to 1400–1600, 2600–3000 and 3600–4400 model years (Supplementary Fig. 7a).

### Climate model validation

Previous studies have extensively assessed the performance of the COSMOS model in simulating the spatial and temporal variations of isotopic composition and hydroclimate changes, demonstrating a strong agreement between model outputs and observational data[43,53,57]. Our model successfully reproduces the characteristic "sawtooth" pattern in the AMOC time series (Fig. 4a). However, it tends to underestimate the magnitude of $\delta^{18}O$ changes over Greenland during the transition from stadial to interstadial (Figs. 3a, 4a). This discrepancy may be attributed to an underestimation of sea-ice responses in the Nordic Seas[53]. In addition, the modeled $\delta^{18}O_p$ changes in the ASM domain are generally smaller than those indicated by proxy records (Figs. 3c, d and 4b, c). Despite these limitations, our model effectively captures the spatial heterogeneity of event contrasts in the ASM domain (Fig. 4d and Table 1), highlighting its capability in representing regional climatic variations.

### Available reconstructions from the Westerlies-dominated region

Marine and lacustrine records, albeit not having sufficient resolution and precision to capture short DO events like DO-15.1, have provided important insights into the interaction between the boreal westerly jet and the ASM at millennial- to multi-millennial timescales. These studies revealed that strong and southward-shifted Westerlies prevailed during cold stadials, inhibiting the EASM, while weak and northward-shifted Westerlies predominated during warm interstadials, prompting the EASM[34,35]. This dynamic alternation was further testified by speleothem trace element records from central China[59]. Our results are consistent with these studies, all suggesting that the boreal Westerlies shifted southward during stadials and northward during interstadials (Figs. 5c and 6).

The proposed dynamic framework invoking the Westerlies in this study is also supported by $\delta^{18}O_c$ records from Hölloch and Kleegruben caves from the Alps, that are indicators of local temperature[82,91], linked qualitatively to meridional shifts of the westerly jet stream. In addition, the Sofular Cave $\delta^{13}C$ record from the Asian Westerlies domain is presumably related to the local temperature[92,93], with a negative excursion corresponding to warmer and wetter climate and thus a northward shift of the Westerlies. Besides, during the last glacial period, Greenland ice-core dust-derived [$Ca^{2+}$] records primarily reflect the latitudinal position and intensity of the Northern Hemisphere westerly winds, as well as the hydroclimate conditions in Asian dust source regions[22,94]. These records lend further support to the notion that the boreal Westerlies shifted northward (southward) during interstadials (stadials), at least qualitatively.

### Modeled cave speleothem $\delta^{18}O$

We also modeled $\delta^{18}O_c$ (Supplementary Fig. 10), which differs from $\delta^{18}O_p$ due to the temperature-related fractionation between calcite and water[17,21]. $\delta^{18}O_c$ was calculated as follows (as same as in ref. 17):

$$\delta^{18}O_{p(VPDB)} = 0.97002 \times \delta^{18}O_{p(SMOW)} - 29.98 \tag{1}$$

$$\delta^{18}O_c = \delta^{18}O_{p(VPDB)} + 2.70 \times 10^6 / T^2 - 3.29. \tag{2}$$

where $T$ is the cave temperature (in Kelvin)[95,96].

The result of the modeled $\delta^{18}O_c$ (Supplementary Fig. 10) is generally consistent with the modeled $\delta^{18}O_p$ (Fig. 4; Supplementary Fig. 6c), indicating that our results are not sensitive to temperature-related fractionation processes.

## Data availability

The U-Th dates for the four speleothems are provided in Supplementary Data 1, the $\delta^{18}O$ time-series for 5 speleothem records are provided in Supplementary Data 2, and other data from the referenced papers shown in the main figures are provided in Supplementary Data 3. The data to support all analyses in this study has been deposited in figshare[97] (https://doi.org/10.6084/m9.figshare.27141303) as well as Zenodo (https://zenodo.org/records/16752697).

## Code availability

The "Mean-fitting" analyses were performed using MATLAB (R2022a). The MATLAB codes used in the "Mean-fitting" analyses are provided in Supplementary Code 1. The standard model code of the 'Community Earth System Models' (COSMOS) version COSMOS-landveg r2413 (2009) is available upon request from the 'Max Planck Institute for Meteorology' in Hamburg (https://www.mpimet.mpg.de). Post-processing of the model output and model data analysis were performed with CDO (Climate Data Operators, version 1.9.5 and 1.9.10, https://code.mpimet.mpg.de/projects/cdo).

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

## Acknowledgements

This work was supported by the National Natural Science Foundation of China (NSFC) grant 42488201 to H.C., and the National Key Research and Development Program of China 2023YFF0805201 to X.Z, NSFC grant 42261144753 to H.Z., NSFC grant 42150710534 to H.C., NSFC grant 41972186 to H.Z., NSFC grant 423B2204 to X.D., and the Strategic Priority Research Program of the Chinese Academy of Sciences (Grant No. XDB 40010203) to Y.C., NSFC grant 42003006 to B.Z. and the grant of State Key Laboratory of Loess Science (SKLLQGZR2401 to H.C.). We thank Qinggang Gao from the University of Melbourne, Xue Jia and Shihao Lei from Xi'an Jiaotong University and Liangcheng Tan and Xiang Mi from the Institute of Earth Environment, Chinese Academy of Sciences, for their help. We thank Yuan Yuan from the National Climate Centre of China for providing permission for using the climate data. We thank the Shennongyuan scenic area for providing support during sample collection.

## Author contributions

H.C., X.Z., and X.D. conceptualized this study. X.D., R.Z., S.H., X.N., Y.N., B.Z., and Y.X. carried out the experiments and data analyses. H.Z., Y.C., D.L., X.W., and F.W.C. conducted the field work. R.Z. and N.M.S. helped organize fieldwork and sampling. X.Z. and Y.Z. analyzed and interpreted model outputs. X.D., X.Z., and H.C. interpreted the results and accomplished the writing. X.D., X.Z., H.Z., S.O.R., Y.C., G.K., C.-P.-M., P.D., A.Svensson, C.S., Y.L., J.W., H.L., A.Sinha, M.W., R.L.E., and H.C. made revisions.

## Competing interests

The authors declare no competing interests.

## Additional information

¹Institute of Global Environmental Change, Xi'an Jiaotong University, Xi'an, China. ²Ice Dynamics and Palaeoclimate, British Antarctic Survey, Cambridge, United Kingdom. ³State Key Laboratory of Tibetan Plateau Earth System, Resources and Environment (TPESRE), Institute of Tibetan Plateau Research, Chinese Academy of Sciences, Beijing, China. ⁴Physics of Ice, Climate and Earth, Niels Bohr Institute, University of Copenhagen, Copenhagen, Denmark. ⁵State Key Laboratory of Loess Science, Institute of Earth Environment, Chinese Academy of Sciences, Xi'an, China. ⁶School of Geography, Nanjing Normal University, Nanjing, China. ⁷Research Institute of Petroleum Exploration & Development, Beijing, China. ⁸Institute of Geology, University of Innsbruck, Innsbruck, Austria. ⁹Earth Observatory of Singapore and Asian School of the Environment, Nanyang Technological University, Singapore, Singapore. ¹⁰Instituto de Geociências, Universidade de São Paulo, São Paulo, Brazil. ¹¹Department of Earth Science, California State University, Carson, CA, USA. ¹²Alfred Wegner Institute, Helmholtz Centre for Polar and Marine Research, Bremerhaven, Germany. ¹³Department of Earth and Environmental Sciences, University of Minnesota, Minneapolis, MN, USA. ¹⁴Faculty of Geography, Yunnan Normal University, Kunming, China. ✉e-mail: xuang@bas.ac.uk; zhanghaiwei@xjtu.edu.cn; cheng021@xjtu.edu.cn

