## [Transparent Peer Review file · Nature Communications]

Interstadial diversity of East Asian summer monsoon linked to changes of the Northern Westerlies

Corresponding Author: Professor Xu Zhang

Version 0:

Reviewer comments:

Reviewer #1

(Remarks to the Author)

The response of tropical and subtropical climates to stadial-interstadial variation remains uncertain. The authors present high-resolution speleothem $\delta^{18}\text{O}$ records from the South American and East Asian monsoon regions, covering two interstadials—one short and one long. Their findings reveal that the East Asian monsoon region exhibits distinct differences in $\delta^{18}\text{O}$ between short and long interstadials, with reduced isotopic depletion during shorter events. In contrast, other regions show consistent isotopic changes regardless of interstadial duration. Using an isotope-enabled climate model, the authors attribute East Asia's unique response to shifts in the latitudinal position of the boreal Westerlies during interstadials. The differences between long and short interstadials are indeed not very well studied in the past. This manuscript is well-written, and I have only a few comments.

Comments:

1. The AMOC self-oscillatory experiment is critical to this study, but it is not sufficiently introduced in the Methods section. While the section includes the model description, readers would benefit from additional experimental details, such as how the experiments were conducted, their duration, and how AMOC overshoot and phase transitions between interstadials and stadials were achieved.
2. While the AMOC experiment is a useful analog for differentiating short and long interstadial phases, it should be acknowledged that there are significant differences from the speleothem records. In the records, long interstadials typically follow stadial phases rather than the “overshoot” observed in the experiment following short interstadials. This distinction is crucial as previous climate states can influence simulation outcomes. Please discuss this difference in the manuscript.
3. Figure 3 illustrates the $\delta^{18}\text{O}$ response in the Asian monsoon region. It would be helpful if the authors also examined the response in Europe and South America, given that speleothem records indicate minimal differences in $\delta^{18}\text{O}$ between short and long interstadials in these regions as well.
4. Although summer precipitation contributes a large portion of annual precipitation in the East Asian monsoon region, autumn and winter precipitation also play significant roles, particularly regarding moisture from the Atlantic Ocean, which tends to be isotopically depleted. Is it possible that reduced Atlantic-sourced moisture, due to the northward shift of the Westerlies, also contributes to the observed $\delta^{18}\text{O}$ enrichment in East Asia?
5. The mechanism proposed—that “in short interstadials, warmer high latitudes push the Westerlies northward”—requires additional support. The authors use the “overshoot” experiment as an analog for short interstadials, with added heat release in the North Atlantic. However, historical short interstadials may not have involved additional high-latitude heat input; their primary distinguishing factor could simply be their duration. This potential disconnect between short interstadials and warmer high latitudes needs to be addressed.

Reviewer #2

(Remarks to the Author)

The paper sets itself out to assess whether the intermittent interstadial warming (D-O events) is translated into tropical climate settings. The rationale for the study is pitched as it is hard to examine as there is limited high-resolution data. The study presents new $\delta^{18}\text{O}$ data from 4 speleothems that are in the South American and East Asian monsoon regions, which have sequences that span a short and long interstadial. These data are coupled with an isotope-enabled climate model, which is used to investigate the drivers of the isotope composition of the speleothems over this interstadial period. Broad conclusions from this work are that the latitudinal position of westerlies influence the $\delta^{18}\text{O}$ of precipitation in Eastern China, while there is no such change in vapour source area in the Indian summer monsoon. The suggestion is that the East Asian summer monsoon has a unique hydroclimate controlled by low and mid-latitude processes that are distinct from the tropics.

General observations and comments:

1. The title of the paper is misleading, as I would have expected a series of interstadials to be investigated, but what this paper actually presents is a very narrow window of a single interstadial between 55 and 57 kyr BP. While they look at two D-O events (15.1; 15.2), which are long and short, this hardly addresses 'diverse' responses of interstadials over the last full glacial. As such, the conclusions they draw are not robust to infer what interstadial responses might be. A title change may solve this, but then it would not speak to the wider scope of Nature Comms.
2. The rationale for why these two specific D-O events is not well developed. I suspect it's because it is based on the level of analysis that could be achieved over this time from these speleothems. However, despite the resolution, the rationale of the 55-57 kyr BP period is not there – are they analogues or typical D-O-type events? This needs to be clearer/more robustly argued.
3. The idea of westerly jet circulation or westerlies controlling monsoon climate is a well-established idea in the literature, so not sure how original the notion westerly movement would be controlling the delivery of moisture to East China and the wider Asian continent.
4. The assertion that summer mid-latitude Westerlies shifted northward during interstadials i.e., when it warms up, allows near-source moisture to be transported, which is $\delta^{18}\text{O}$ -enriched, is a different mechanism than that usually proposed for controlling speleothem $\delta^{18}\text{O}$ in China, so does that mean those records are also controlled by vapour source changes rather than the monsoon directly?
5. Discussions around the role of the westerlies and D-O-type events have been established from the Japan Sea. Admittedly, not at the same resolution, but this is important literature that considers this connection over long timescales, yet it is absent from any of the discussion.
6. It's especially clear why the Brazilian speleothem is included. Yes, it's a counter point to the changes in the northern Hemisphere, but the record isn't unpacked in anyway and rather sits there. Also, it can only shed light on DO-15.1, so what is its value in the MS. The data from Europe and India make sense to develop the westerly argument, have something located far field isn't very well integrated.
7. The discussion on the ISM, SASM and EASM is rather throwaway and doesn't consider the vast body of literature out there. To simply label them as just tropical vs. sub-tropical dominated is oversimplifying the modern and past climatology.
8. There are elements of circulatory in the analysis of the data because they are tuned to the GICC05 chronology. The methods outline the proxies used, but surely if you are going to tune a record then comparable proxies should be used, i.e. dust for dust, for acknowledge that is now an accepted approach.
9. The difference between the records is only an amplitude of 0.5 per mille, given that you have an isotope enabled model, sure you can quantify the difference in vapor source to drive this?
10. I found difficult to follow your logic with regards to the AMOC overshoot and interstadial phases. Is the model not sensitive enough to reproduce the short vs long DOs? It could just be my misunderstanding, but some clarity would be good.
11. The literature around the land-sea contrast has been significantly updated. It invokes the Himalayas as a block to westerly flow, so probably important in your thinking around connections to the North Atlantic. The idea of westerly jet circulation is nothing new see the Chiang et al., papers about this (included in your ref list).
12. When discussing the transport of water vapor from the western Pacific to the EASM region, how much more water vapor would there? Can you calculate this from the model? What about changes in the isotopic composition of the sea water? Does that change?
13. You state that the water vapor in the ISM is remotely sourced from the Indian Ocean, are you suggesting it is coming from the Southern Hemisphere? Modern climatology shows that vapor is very much locally sourced from the west coast of India in the Arabian Sea and from the Bay of Bengal.

Version 1:

Reviewer comments:

Reviewer #1

(Remarks to the Author)

I appreciate the authors' effort in revising the manuscript. My previous concerns have been well addressed. I have no further comments.

Reviewer #3

(Remarks to the Author)

This manuscript presents a very high-resolution $\delta^{18}\text{O}$ record from four stalagmites in the Indian and East Asian monsoon regions, covering a brief interstadial period with precise age control. I agree that this dataset is valuable for deepening our understanding of spatial hydrological variability during interstadials.

However, in the revised manuscript, the central scientific questions remain unclear. For the study to engage a broader readership of this journal, the authors should state more clearly their key motivations and implications of their work:

1. Why is it important to investigate climate variability in tropical and subtropical regions during the interstadials? Simply citing the lack of high-resolution data may not be sufficient.

2. In what context do the authors classify interstadials based on their duration and try to find the difference among them? Could the results of this study offer new insights into the length of interstadials and their underlying mechanisms?

3. Why is the result of a northward westerly jet position during this short interstadial notable, especially when northward patterns have been previously reported for longer interstadials?

I think that clarifying these scientific motivations and their implications would strengthen the manuscript.

Moderate concerns:

Lines 68-70: Are there any other short interstadials, around 100 years in duration, besides DO-15.1?

Lines 84-91: The significance of addressing these knowledge gaps is not sufficiently explained.

Lines 102-106: The current description may cause misunderstanding. It might be already established that the westerlies shift northward during interstadials compared to stadials. Therefore, the authors should clarify that what they observe during the short interstadial is a further northward shift compared to the shift observed during longer interstadials.

Lines 216-225: I still wonder why the AMOC overshoot at the onset of interstadials is considered an appropriate analogue for short interstadials, simply because of the similar $\delta^{18}\text{O}$ values observed during the overshoot phase and during D-O 15.1 in both Greenland ice cores and stalagmite records from the EASM region. Are the mechanisms that generate the overshoot and those responsible for the short interstadial comparable?

Figure 4d: I thought that many readers, including myself, may find it difficult to grasp the meaning of the ratio ' $\delta^{18}\text{O}_p$ (Interstadial minus Overshoot phases) / (Interstadial minus Stadial phases)'. Could the authors consider presenting this trend in a more straightforward way to better illustrate the $\delta^{18}\text{O}$ evolution during the overshoot phase?

Version 2:

Reviewer comments:

Reviewer #3

(Remarks to the Author)

The authors have carefully revised the manuscript and provided clear responses to the reviewers' comments.

I believe the revised version meets the journal's standards, and I have no further concerns.

I support the publication of this manuscript.

**A point-by-point response to the reviews**

(Original comments are in *blue*, and our responses are in *black*)

**1. Comments from Reviewer #1:**

*General remark.*

The response of tropical and subtropical climates to stadial-interstadial variation
remains uncertain. The authors present high-resolution speleothem $\delta^{18}\text{O}$ records from
the South American and East Asian monsoon regions, covering two interstadials—one
short and one long. Their findings reveal that the East Asian monsoon region exhibits
distinct differences in $\delta^{18}\text{O}$ between short and long interstadials, with reduced isotopic
depletion during shorter events. In contrast, other regions show consistent isotopic
changes regardless of interstadial duration. Using an isotope-enabled climate model,
the authors attribute East Asia's unique response to shifts in the latitudinal position of
the boreal Westerlies during interstadials. The differences between long and short
interstadials are indeed not very well studied in the past. This manuscript is
well-written, and I have only a few comments.

**Response:** We sincerely thank the reviewer for his/her positive feedback on our
manuscript, which helped to improve overall quality of our work. Below, we provide
detailed responses to the specific comments raised by the reviewer.

*Comment 1.*

The AMOC self-oscillatory experiment is critical to this study, but it is not sufficiently
introduced in the Methods section. While the section includes the model description,
readers would benefit from additional experimental details, such as how the
experiments were conducted, their duration, and how AMOC overshoot and phase
transitions between interstadials and stadials were achieved.

**Response:** This is a very good suggestion, and accordingly we have added more
information on the experimental details in the Methods section.

**Climate model description**

*"Model settings. We used a comprehensive fully coupled atmosphere–ocean general*
*circulation model (AOGCM), COSMOS (ECHAM5-JSBACH-MPI-OM) in this study.*
*The atmospheric model ECHAM5 (Roeckner et al., 2003), complemented by the land*
*surface component JSBACH (Brovkin et al., 2009), was used at T31 resolution*
*($\sim 3.75^\circ$), with 19 vertical layers. The ocean model MPI-OM (Marsland et al., 2003),*
*including sea-ice dynamics formulated using viscous-plastic rheology (Hibler, 1979),*

has a resolution of GR30 ($3^{\circ} \times 1.8^{\circ}$) in the horizontal, with 40 uneven vertical layers.
Water isotopes have been implemented in COSMOS to enable a direct comparison
between simulated and observed isotopic values (Werner et al., 2016). In this study,
we employed the equilibrium experiment (E40ka_34Orb) with ice volume and
atmospheric CO₂ concentrations (195 ppm) fixed at an equilibrated control
simulation (E40ka_CTL), while the orbital settings were set to 34 ka BP (Zhang et al.,
2021). The specific orbital parameters are as follow: the eccentricity is 0.014996, the
precession of the equinoxes (the angle between the Earth 's position during the
Northern Hemisphere vernal equinox and the orbit perihelion) is 84.84°, and the
obliquity is 22.6°. The model was integrated
for 5,500 years to explore the equilibrated responses of the glacial climate to orbital
configurations at 34 ka BP (Zhang et al., 2021). Notably, 34 ka BP is a time when
internal climate background conditions (ice volume (Spratt & Lisiecki, 2016) and
atmospheric CO₂ level (Bauska et al., 2021)) are relatively constant and similar to
that at 40 ka BP.

The E40ka_34Orb experiment is characteristic of millennial-scale AMOC
self-oscillation, resembling recorded DO cycles during MIS3 (Zhang et al., 2021) (Fig.
3a). Overall, the COSMOS model has already been used to investigate a range of
paleoclimate phenomena, especially millennial-scale abrupt glacial climate changes
such as DO events (Knorr et al., 2021; Maier et al., 2018; Zhang et al., 2014, 2017,
2021), therefore it is a suitable model for this study.

**Definition of different phases.** In this study, the modelled AMOC index (Fig. 4a) is
defined by the maximum stream function between 500–2500 m in the North Atlantic
north of 45°N. We classified three AMOC phases: Stadial phases (900–1,050,
2,150–2,350 and 3,300–3,350 model years), Overshoot phases (1,150–1,200,
2,450–2,500 and 3,480–3,550 model years) and Interstadial phases (1,150–1,650,
2,450–3,000 and 3,480–4,400 model years) (Fig. 4a). In our simulations, the long
interstadials were specifically designed to incorporate the overshoot phases, a feature
consistent with observations of most long interstadials in Greenland (Fig. 3a, b).

To further investigate whether the presence of the "overshoot" phase might
influence our conclusions, an alternative "interstadial" phase by excluding the prior
"overshoot" phase is defined. In this scenario, the stadial and overshoot phases
remain unchanged, and the interstadial phases changed to 1,400–1,600, 2600–3,000
and 3,600–4,400 model years (Supplementary Fig. 7a)."

**Comment 2.**

While the AMOC experiment is a useful analog for differentiating short and long

interstadial phases, it should be acknowledged that there are significant differences
from the speleothem records. In the records, long interstadials typically follow stadial
phases rather than the “overshoot” observed in the experiment following short
interstadials. This distinction is crucial as previous climate states can influence
simulation outcomes. Please discuss this difference in the manuscript.

**Response:** We appreciate the reviewer's insightful comment regarding the potential
influence of previous climate states on simulation outcomes. While we acknowledge
that such factors can indeed play a role, we would like to clarify that the mechanism
explored in this study is independent of the selection of scenarios. In our simulations,
the long interstadials were specifically designed to incorporate the overshoot phases
(Fig. 4a), a feature consistent with observations of most long interstadials in
Greenland. These interstadials typically exhibit a "sawtooth" structure, characterized
by a brief overshoot phase followed by a relatively stable condition, as documented in
NGRIP and GISP2 ice-core $\delta^{18}\text{O}$ records spanning DO-3 to DO-14 (Dansgaard et al.,
1993; Grootes et al., 1993; Johnsen et al., 1992; Rasmussen et al., 2014) (Fig. 3a, b).
Our definition of the Interstadial phases in our experiments enables us to effectively
analyze the differences between long and short DO interstadials.

To further investigate whether the presence of the "overshoot" phase might
influence our conclusions, we defined interstadial phases excluding the overshoot
phases (Supplementary Fig. 7a; Methods), which shows similar results as our earlier
findings. These results coherently show minimal difference in India and pronounced
differences in Southeast China between the long and short interstadials (Fig. 4d;
Supplementary Fig. 7c). This contrast is primarily driven by warmer high-latitude
conditions during the short interstadials (Fig. 5d; Supplementary Fig. 7e), which
favors a northward shift of the interstadial Northern Westerlies (Fig. 5c, d;
Supplementary Fig. 7d). This facilitates import of isotopically-enriched near-source
water vapor from subtropical western Pacific to Southeast China during the short
interstadials (Fig. 4b, c; Supplementary Fig. 7b). These results further strengthen the
reliability of our conclusions.

*Comment 3.*

Figure 3 illustrates the $\delta^{18}\text{O}$ response in the Asian monsoon region. It would be
helpful if the authors also examined the response in Europe and South America, given
that speleothem records indicate minimal differences in $\delta^{18}\text{O}$ between short and long
interstadials in these regions as well.

**Response:** We sincerely appreciate the reviewer's insightful comment. The variations
in $\delta^{18}\text{O}$ of precipitation within the South American summer monsoon (SASM) domain

are illustrated in Fig. R1. Currently, our speleothem record is the sole dataset that
 encompasses both the short DO event (DO-15.1) and the long DO events in the
 SASM domain. As a result, it is challenging to draw unequivocal conclusions
 regarding the spatial patterns across the vast SASM domain, underscoring the
 necessity for additional high-resolution reconstructions. In the revised manuscript, we
 have shifted our focus primarily to the ASM system, reducing the detailed discussion
 of the SASM region. However, we have retained the South American speleothem
 record as independent evidence to support the role of AMOC changes during DO-15.1.
 This record provides critical insights into the relationship between AMOC dynamics
 and tropical hydroclimate variability, particularly through its clear anti-phase
 relationship with Asian speleothem $\delta^{18}\text{O}$ records, which aligns with the "monsoon
 seesaw" pattern (Line 178–Line 188 in the revised manuscript).

**Response Figure 1. Simulated $\delta^{18}\text{O}$ changes in South America during DO warmings.**

Modelled changes in mean annual precipitation $\delta^{18}\text{O}$ ($\delta^{18}\text{O}_p$) (units: ‰) during (a) short
 (Overshoot minus Stadial phases) and (c) long (Interstadial minus Stadial phases) warming
 events. (b) Simulated changes between long and short interstadials (Interstadial minus
 Overshoot phases). The black dot shows the cave location. The time periods of simulated
 Stadial, Overshoot and Interstadial phases refer to Fig. 4a.

Two European speleothem records encompass the DO-15.1 event (Supplementary
 Fig. 5g, h). The overall trends in $\delta^{18}\text{O}$ variations between interstadials and stadials
 align with our model predictions (Fig. R2a, c). Nonetheless, while the timing of
 climate events is consistent across these records (Supplementary Fig. 5g, h) and the
 two caves are geographically proximate (Fig. R2b), there is a discrepancy in the
 recorded amplitude differences between DO-15.1 and the longer DO events
 (Supplementary Fig. 5g, h). Consequently, our analyses for this region remain
 preliminary at this stage, and additional records are essential to accurately delineate
 the specific characteristics of these events in the region.

**Response Figure 2. Simulated $\delta^{18}\text{O}$ changes in Europe during DO warmings.** Modelled
 changes in mean annual precipitation $\delta^{18}\text{O}$ ($\delta^{18}\text{O}_p$) (units: ‰) during (a) short (Overshoot
 minus Stadial phases) and (c) long (Interstadial minus Stadial phases) warming events. (b)
 Simulated changes between long and short interstadials (Interstadial minus Overshoot phases).
 The black dots show the cave locations. The time periods of simulated Stadial, Overshoot and
 Interstadial phases refer to Fig. 4a.

*Comment 4.*

Although summer precipitation contributes a large portion of annual precipitation in
 the East Asian monsoon region, autumn and winter precipitation also play significant
 roles, particularly regarding moisture from the Atlantic Ocean, which tends to be
 isotopically depleted. Is it possible that reduced Atlantic-sourced moisture, due to the
 northward shift of the Westerlies, also contributes to the observed $\delta^{18}\text{O}$ enrichment in
 East Asia?

**Response:** This is a very good question. Moisture from the Atlantic Ocean,
 Mediterranean, etc. play important roles in the Westerlies Asia (Cheng et al., 2012,
 2016; Kutzbach et al., 2014; Tan et al., 2024). However, based on simulated moisture
 transports in different seasons during both the interstadial and the overshoot phases,
 contribution of the Atlantic-sourced moisture to monsoonal Asia is limited (Fig. R3).

**Response Figure 3. Modelled vertical integrated water-vapor transport (WT) (vectors, units:**

159 kg/m/s) between long and short interstadials (Interstadial minus Overshoot phases). DJF:
December-January-February; MAM: March-April-May; JJA: June-July-August; SON:
September-October-November. The time periods of simulated Stadial, Overshoot and
Interstadial phases refer to Fig. 4a.

*Comment 5.*

The mechanism proposed—that “in short interstadials, warmer high latitudes push the
Westerlies northward”—requires additional support. The authors use the “overshoot”
experiment as an analog for short interstadials, with added heat release in the North
Atlantic. However, historical short interstadials may not have involved additional
high-latitude heat input; their primary distinguishing factor could simply be their
duration. This potential disconnect between short interstadials and warmer high
latitudes needs to be addressed.

**Response:** We thank the reviewer for his/her insightful comment and would like to
provide additional evidence to support our proposed mechanism— that is, the
intensity of high-latitude warming, rather than the duration of interstadials, is the
essential factor driving the northward shift of the Northern Westerlies during
interstadials.

First, the warmer high-latitude conditions during DO-15.1 and the "overshoot"
phases of long interstadials are well characterized by Greenland ice-core $\delta^{18}\text{O}$ records.
These records are widely used as a local surface temperature proxy (e.g., NGRIP
Project Members, 2004; Gkinis et al., 2014), demonstrate that DO-15.1 and the
overshoot phases of long interstadials were warmer than the subsequent interstadial
phases (Fig. 3a, b). This provides direct evidence of warmer high-latitude conditions
(at least over Greenland) during these overshoot phases.

Second, although existing proxy records used to infer changes in the boreal
Westerlies are qualitative and lack the resolution and precision to robustly disentangle
changes during DO-15.1, they consistently suggest a northward (southward) shift of
the Asian Westerlies during interstadial (stadial) periods (An et al., 2012; Chiang et al.,
2015; Nagashima et al., 2011; Zhang et al., 2018), consistent with our modelling
results (Supplementary Fig. 6i). To further test our hypothesis, we examined the
spatial heterogeneity in the structure of long interstadials in the ASM domain. If our
hypothesis is correct, we would expect considerably less depleted $\delta^{18}\text{O}$ values over
Southeast China during the overshoot phases of long interstadials, reflecting a more
pronounced northward shift of the Westerlies.

The Xianyun cave record from Southeast China provides evidence supporting this
expectation, as it demonstrates that the $\delta^{18}\text{O}$ values during the overshoot phases of

DO-8, DO-12, and DO-14 are less depleted than those observed in their respective
subsequent periods (Fig. 3c). In contrast, the Bittoo cave speleothem $\delta^{18}\text{O}$ record from
the ISM domain shows minor differences between the overshoot and subsequent
interstadial phases (Fig. 3d). These findings align well with our proposed mechanism,
providing additional support for the robustness of our conclusions. This suggests that
our mechanism has broad applicability: it not only explains the differences between
long and short interstadials, but also aids in interpreting the relatively gradual
depletion trend of speleothem $\delta^{18}\text{O}$ values in Xianyun cave during long interstadials,
compared with those in Bittoo cave.

**Reference:**

- An, Z., Colman, S., Zhou, W. et al. Interplay between the Westerlies and Asian monsoon recorded
in Lake Qinghai sediments since 32 ka. *Sci Rep* **2**, 619 (2012).
- Bauska, T. K., Marcott, S. A. & Brook, E. J. Abrupt changes in the global carbon cycle during the
last glacial period. *Nat. Geosci.* **14**, 91-96 (2021).
- Brovkin, V., Raddatz, T., Reick, C. H., Claussen, M. & Gayler, V. Global biogeophysical
interactions between forest and climate. *Geophys. Res. Lett.* **36**, L07405 (2009).
- Cheng, H., Sinha, A., Wang, X. et al. The Global Paleomonsoon as seen through speleothem
records from Asia and the Americas. *Clim Dyn* **39**, 1045–1062 (2012).
- Cheng, H., Spötl, C., Breitenbach, S. et al. Climate variations of Central Asia on orbital to
millennial timescales. *Sci Rep* **6**, 36975 (2016).
- Chiang, J. C. H. et al. Role of seasonal transitions and westerly jets in East Asian paleoclimate.
*Quat. Sci. Rev.* **108**, 111-129 (2015).
- Dansgaard, W. et al. Evidence for general instability of past climate from a 250-kyr ice-core
record. *Nature* **364**, 218-220 (1993).
- Johnsen, S. J. et al. The $\delta^{18}\text{O}$ record along the Greenland Ice Core Project deep ice core and the
problem of possible Eemian climatic instability. *J. Geophys. Res.-Oceans* **102** (1997).
- Gkinis, V., Simonsen, S. B., Buchardt, S. L., White, J. W. C. & Vinther, B. M. Water isotope
diffusion rates from the NorthGRIP ice core for the last 16,000 years – Glaciological and
paleoclimatic implications. *Earth Planet. Sci. Lett.* **405**, 132-141 (2014).
- Grootes, P. M. & Stuiver, M. Oxygen 18/16 variability in Greenland snow and ice with 10^{-3} - to
10^5 - year time resolution. *J. Geophys. Res.-Oceans* **102**, 26455-26470 (1997).
- Hibler, W. D. A Dynamic Thermodynamic Sea Ice Model. *J. Phys. Oceanogr.* **9**, 815-846 (1979).
- Knorr, G. et al. A salty deep ocean as a prerequisite for glacial termination. *Nat. Geosci.* **14**,
930-936 (2021).
- Kutzbach, J.E., Chen, G., Cheng, H. et al. Potential role of winter rainfall in explaining increased
moisture in the Mediterranean and Middle East during periods of maximum orbitally-forced
insolation seasonality. *Clim Dyn* **42**, 1079–1095 (2014).
- Maier, E. et al. North Pacific freshwater events linked to changes in glacial ocean circulation.

*Nature* **559**, 241-245 (2018).

Marsland, S. J., Haak, H., Jungclaus, J. H., Latif, M. & Röske, F. The Max-Planck-Institute global
ocean/sea ice model with orthogonal curvilinear coordinates. *Ocean Model.* **5**, 91-127 (2003).

Nagashima, K. et al. Millennial-scale oscillations of the westerly jet path during the last glacial
period. *J. Asian Earth Sci.* **40**, 1214-1220 (2011).

North Greenland Ice Core Project Members. High-resolution record of Northern Hemisphere
climate extending into the last interglacial period. *Nature* **431**, 147-151 (2004).

Rasmussen, S. O. et al. A stratigraphic framework for abrupt climatic changes during the Last
Glacial period based on three synchronized Greenland ice-core records: refining and
extending the INTIMATE event stratigraphy. *Quat. Sci. Rev.* **106**, 14-28 (2014).

Roeckner, E. et al. The Atmospheric General Circulation Model ECHAM5. Part 1: Model
Description (Max-Planck-Institut für Meteorologie, 2003).

Spratt, R. M. and Lisiecki, L. E.: A Late Pleistocene sea level stack. *Clim. Past* **12**, 1079–1092
(2016).

Tan, L. et al. Hydroclimatic changes on multiple timescales since 7800 y BP in the winter
precipitation–dominated Central Asia. *Proc. Natl. Acad. Sci. USA* **121**, e2321645121 (2024).

Werner, M. et al. Glacial–interglacial changes in H₂¹⁸O, HDO and deuterium excess – results from
the fully coupled ECHAM5/MPI-OM Earth system model. *Geosci. Model Dev.* **9**, 647-670
(2016).

Zhang, H. et al. East Asian hydroclimate modulated by the position of the westerlies during
Termination I. *Science* **362**, 580-583 (2018).

Zhang, X., Lohmann, G., Knorr, G. & Purcell, C. Abrupt glacial climate shifts controlled by ice
sheet changes. *Nature* **512**, 290-294 (2014).

Zhang, X., Knorr, G., Lohmann, G. et al. Abrupt North Atlantic circulation changes in response to
gradual CO₂ forcing in a glacial climate state. *Nature Geosci* **10**, 518–523 (2017).

Zhang, X. et al. Direct astronomical influence on abrupt climate variability. *Nature Geosci* **14**,
819-826 (2021).

**2. Comments from Reviewer #2:**

***General remark.***

The paper sets itself out to assess whether the intermittent interstadial warming (D-O
events) is translated into tropical climate settings. The rationale for the study is
pitched as it is hard to examine as there is limited high-resolution data. The study
presents new $\delta^{18}\text{O}$ data from 4 speleothems that are in the South American and East
Asian monsoon regions, which have sequences that span a short and long interstadial.
These data are coupled with an isotope-enabled climate model, which is used to
investigate the drivers of the isotope composition of the speleothems over this
interstadial period. Broad conclusions from this work are that the latitudinal position

of westerlies influence the ^{18}O of precipitation in Eastern China, while there is no
such change in vapour source area in the Indian summer monsoon. The suggestion is
that the East Asian summer monsoon has a unique hydroclimate controlled by low
and mid-latitude processes that are distinct from the tropics.

**Response:** We sincerely appreciate the reviewer's insightful comments, which have
greatly contributed to the refinement of our manuscript. In response to the reviewer's
comments, we have taken three steps to strengthen our study. First, we have provided
additional evidence to support our conclusions, as detailed in the responses below.
Second, we have constructed a new high-resolution BT-1 record from Bittoo Cave in
the Indian summer monsoon (ISM) domain, incorporating approximately 880 new
oxygen-carbon isotope measurements. This new dataset reinforces our conclusion
regarding the distinct hydroclimate responses in different monsoon regions.
Additionally, we have expanded our discussion to include all Dansgaard-Oeschger
(DO) events during Marine Isotope Stage 3 (MIS3), thereby broadening the scope and
significance of our study. The reviewer's feedback has been invaluable in enhancing
the clarity and depth of our manuscript, and we are grateful for his/her constructive
input.

*Comment 1.*

The title of the paper is misleading, as I would have expected a series of interstadials
to be investigated, but what this paper actual presents is a very narrow window of a
single interstadial between 55 and 57 kyr BP. While they look at two D-O events
(15.1; 15.2), which are long and short, this hardly addresses 'diverse' responses of
interstadials over the last full glacial. As such, the conclusions they draw are not
robust to infer what interstadial responses might be. A title change may solve this, but
then it would not speak to the wider scope of Nature Comms.

**Response:** We thank the reviewer for this valuable comment. We agree that focusing
solely on the comparison between DO-15.1 and DO-15.2 represents a relatively
narrow perspective. In response, we have significantly expanded our discussion in the
revised manuscript to include all DO events that occurred during MIS3. This broader
analysis allows us to compare a wider range of DO events (Fig. 3), resulting in a more
comprehensive understanding of the climatic variations during this period.

Firstly, the comparison of various DO events during MIS3 reveals that the
amplitude of $\delta^{18}\text{O}$ depletion in Southeast China during DO-15.1 is smaller than that
observed during longer DO events (Fig. 3c; Table 1). In contrast, the amplitude of
$\delta^{18}\text{O}$ changes in the Bittoo cave record (ISM domain) during short DO event is
comparable to the changes observed in longer DO events (Fig. 3d; Table 1). These

observations are consistent with our previous findings.

Secondly, existing proxy records used to infer changes in the boreal Westerlies
consistently suggest a northward (southward) shift of the Asian Westerlies during
interstadial (stadial) periods (An et al., 2012; Chiang et al., 2015; Nagashima et al.,
2011; Zhang et al., 2018), which agree with our simulations (Fig. 5c; Supplementary
Fig. 6i).

Thirdly, cave speleothem $\delta^{18}\text{O}$ manifests observable spatial heterogeneity in the
vast ASM domain (Fig. 1; Supplementary Fig. 5), which is captured by our model
simulations (Fig. 4d), providing further support for the reliability of our model (Line
276–Line 293 in the revised manuscript).

Additionally, we have used other DO events to further test our hypothesis. If our
hypothesis is correct, we would expect considerably less depleted $\delta^{18}\text{O}$ values over
Southeast China during the overshoot phases of long interstadials, reflecting a more
pronounced northward shift of the Westerlies. The Xianyun cave record from
Southeast China provides evidence supporting this expectation, as it demonstrates that
the $\delta^{18}\text{O}$ values during the overshoot phases of DO-8, DO-12, and DO-14 are less
depleted than those observed in their respective subsequent periods (Fig. 3c). In
contrast, the Bittoo cave speleothem $\delta^{18}\text{O}$ record from the ISM domain shows minor
differences between the overshoot and subsequent interstadial phases (Fig. 3d). These
findings align well with our proposed mechanism, providing additional support for the
robustness of our conclusions. This suggests that our mechanism has broad
applicability: it not only explains the differences between long and short interstadials,
but also aids in interpreting the relatively gradual depletion trend of speleothem $\delta^{18}\text{O}$
values in Xianyun cave during long interstadials, compared with those in Bittoo cave.

These refinements not only strengthen the scientific rigor of our study but also
ensures that our manuscript aligns more closely with its title and meets the
expectations of the readers of *Nature Communications*. We believe these changes
have substantially improved the scope and impact of our work.

*Comment 2.*

The rationale for why these two specific D-O events is not well developed. I suspect
it's because it is based on the level of analysis that could be achieved over this time
from these speleothems. However, despite the resolution, the rationale of the 55-57
342 kyr BP period is bot there – are they analogues or typical D-O-type events? This
needs to be clearer/more robustly argued.

**Response:** Thank you for the comment. In the revised version of our manuscript, we
inspected all DO events that occurred throughout the MIS3 (Fig. 3; Table 1), and our

conclusion still holds, as discussed in our response to Comment 1. This resolves the
issue on the representativeness of DO events.

*Comment 3.*

The idea of westerly jet circulation or westerlies controlling monsoon climate is a
well-established idea in the literature, so not sure how original the notion westerly
movement would be controlling the delivery of moisture to East China and the wider
Asian continent.

**Response:** We sincerely thank the reviewer for raising this important point. The role
of the meridional shift of the Northern Westerlies in modulating the EASM is indeed a
well-documented concept in the literature (e.g., Chiang et al., 2015; Zhang et al.,
2018). Building on this foundation, our isotope-enabled model provides direct
evidence to support this idea, demonstrating that the Westerlies shift northward during
both long and short interstadials, while moving southward during stadial periods (Fig.
5c; Supplementary Fig. 6i). More importantly, our study advances the existing
understanding by highlighting that the duration of interstadials (i.e., the intensity of
warming in northern high latitudes) also plays a critical role in shaping precipitation
$\delta^{18}\text{O}$ variability in the EASM region, further elucidating the influence of the
Westerlies in this process. Below, we elaborate on the novel contributions of our work
in this context.

Modern-day studies have extensively explored the interaction between the
westerly jet and the monsoon climate on annual to inter-annual timescales (e.g.,
Chiang et al., 2020; Schiemann et al., 2009), paleoclimate investigations into the role
of the westerly jet at millennial timescales remain relatively limited. Previous
paleoclimate studies have primarily focused on essential contrasts between stadials
and interstadials, such as the southward shift of the Westerlies and weakened Asian
Summer Monsoon (ASM) during stadials, versus a northward shift and strengthened
ASM during interstadials (e.g., Nagashima et al., 2011; Chiang et al., 2015; Zhang et
al., 2018). However, the specific role of the Westerlies and associated moisture
transport during different types of interstadials (e.g., short vs. long) during the last
glacial period remains poorly understood, largely due to the scarcity of
high-resolution records and pertinent model scrutinization (e.g., with water isotopes).

Our study fills this gap by presenting new high-resolution speleothem records and
employing an isotope-enabled climate model to scrutinize the details. This approach
allows us to disentangle the distinct influence of the Westerlies during different
interstadials, thus providing new insights into the mechanisms driving hydroclimate
variability in East Asia during millennial-scale climate events. We believe this

represents a significant advancement in understanding the interplay between the
Westerlies and monsoon systems on millennial-centennial timescales, particularly
during the last glacial period.

*Comment 4.*

The assertion that summer mid-latitude Westerlies shifted northward during
interstadials i.e., when it warms up, allows near-source moisture to be transported,
which is ^{18}O -enriched, is a different mechanism than that usually proposed for
controlling speleothem $\delta^{18}\text{O}$ in China, so does that mean those records are also
controlled by vapour source changes rather than the monsoon directly?

**Response:** We sincerely appreciate the reviewer's insightful comment, which allows
394 us to further clarify the mechanisms controlling speleothem $\delta^{18}\text{O}$ in the Asian
Summer Monsoon (ASM) region and the relationship between vapor source changes
and monsoon dynamics.

First and foremost, it is crucial to highlight that the discrepancies observed
between the ISM and EASM records in this study reflect a subtle yet significant
distinction. However, the primary characteristic of DO events remains consistent
across both ISM and EASM records, with higher $\delta^{18}\text{O}$ values during stadials and
lower $\delta^{18}\text{O}$ values during interstadials (Cai et al., 2006; Cheng et al., 2019; Wang et
al., 2001) (Fig. 6b–d). This uniformity underscores the robust and coherent response
of the ASM system to global climate variability on millennial timescales.

Regarding the reviewer's question about whether speleothem $\delta^{18}\text{O}$ records in
China are controlled by vapor source changes rather than the monsoon directly, we
would like to clarify that these two mechanisms are not mutually exclusive. While
speleothem $\delta^{18}\text{O}$ in the ASM domain has traditionally been interpreted as a proxy for
monsoon strength, recent advances in proxy records, modern monitoring, and
numerical modeling simulations have demonstrated that ASM speleothem $\delta^{18}\text{O}$ at
multi-timescales primarily reflects large-scale atmospheric circulation and the
meridional shift of tropical rain belts. These changes are driven by variations in vapor
source regions and upstream convection (or rainout effect along the trajectory of
moisture transport), rather than local rainfall amount alone (Cai et al., 2018; Cheng et
al., 2019, 2021a, 2021b; He et al., 2021; Hu et al., 2019; Kathayat et al., 2021; Liu et
al., 2020; Tabor et al., 2018; Zhang et al., 2021; Zhao et al., 2023) (Supplementary
Note 1.1).

The northward shift of the mid-latitude Westerlies during interstadials, as
proposed in our study, facilitates the transport of near-source, $\delta^{18}\text{O}$ -enriched
moisture/water vapor to the EASM region. This mechanism aligns with the

established understanding that changes in vapor source regions and upstream
convection are integral to the interpretation of speleothem $\delta^{18}\text{O}$ records. A weaker
large-scale atmospheric circulation would naturally involve a reduced contribution
from remote moisture sources, consistent with the findings of previous studies (e.g.,
Cai et al., 2018; Hu et al., 2019). Therefore, our proposed mechanism does not
contradict the existing interpretation framework, but rather complements it by
highlighting the role of Westerlies in modulating the regional isotopic composition via
amendments of the monsoon circulation and the associated moisture transport on
millennial timescales.

In sum, speleothem $\delta^{18}\text{O}$ records in the ASM region are influenced by
multifaceted processes in the monsoon dynamic system, including changes in vapor
source regions, upstream convections, and large-scale atmospheric circulations etc.
Our findings provide additional insights into the complexity of the interplays among
these processes during DO events. We hope that this clarification addresses the
reviewer's concern and aids in understanding our results.

***Comment 5.***

Discussions around the role of the westerlies and D-O-type events have been
established from the Japan Sea. Admittedly, not at the same resolution, but this is
important literature that considers this connection over long timescales, yet it is absent
from any of the discussion.

**Response:** Thanks for the suggestion. The marine sediment records from the Sea of
Japan have been used to probe the role of the Westerlies across the prolonged DO
events, which indeed provide important results regarding the connection between the
Westerlies and monsoon on the long-term timescales. In the revised manuscript, we
not only incorporated the literature mentioned by the reviewer, but also discussed
other relevant climate records and model results between Line 565 and Line 574:

*“Marine and lacustrine records, albeit not having sufficient resolution and precision*
*to capture short DO events like DO-15.1, have provided important insights into the*
*interaction between the boreal westerly jet and the ASM at the millennial- to*
*multi-millennial timescales. These studies revealed that the strong and*
*southward-shifted Westerlies prevailed during cold stadials, inhibiting the EASM,*
*while a weak and northward-shifted Westerlies predominated during warm*
*interstadials, prompting the EASM (An et al., 2012; Nagashima et al., 2011). This*
*dynamic alternation was further testified by speleothem trace element records from*
*central China (Zhang et al., 2018) and the climate model simulations (Chiang et al.,*
*2015). Our results reported here is consistent with these studies, all suggesting that*

*the boreal Westerlies shifted southward during stadials and northward during*
*interstadials (Figs. 5c, 6; Supplementary Fig. 6i).”*

**Comment 6.**

It’s especially clear why the Brazilian speleothem is included. Yes, it’s a counter point
to the changes in the northern Hemisphere, but the record isn’t unpacked in anyway
and rather sits there. Also, it can only shed light on DO-15.1, so what is its value in
the MS. The data from Europe and India make sense to develop the westerly argument,
have something located far field isn’t very well integrated.

**Response:** We understand the reviewer's comments and agree that the role of the
Brazilian speleothem record in the manuscript needs further clarification. Currently,
our speleothem record is the only dataset in the South American Summer Monsoon
(SASM) region that covers both the short DO event (DO-15.1) and the long DO
events. However, due to the limited data available in the SASM region, it is
challenging to draw unequivocal conclusions about the spatial patterns across the
entire SASM domain or to conduct a detailed comparison of isotopic variations with
the EASM. Therefore, in the revised manuscript, we have focused our discussion
primarily on the differences between the ISM and the EASM. Nevertheless, we have
retained the South American speleothem record as independent evidence to support
the role of AMOC changes during DO-15.1. The Brazilian speleothem $\delta^{18}\text{O}$ record
shows a clear anti-phase relationship with the Asian speleothem $\delta^{18}\text{O}$ records (Fig.
2c–g), reflecting the "monsoon seesaw" pattern, which supports the hypothesis of
ITCZ shifts driven by large-scale AMOC changes (Line 178–Line 188 in the revised
manuscript).

On the other hand, the European records help to develop the Westerlies argument,
since they reflect to some extent the North Atlantic storm tracks, i.e., the boreal
Westerlies position in Europe. Accordingly, we added the following discussion in the
revised manuscript between Line 575 and Line 586:

*“The proposed dynamic framework invoking the Westerlies in this study is also*
*supported by speleothem $\delta^{18}\text{O}$ records from Hölloch and Kleegruben caves in the Alps*
*(Moseley et al., 2014; Spötl et al., 2006), that are indicators of local temperature,*
*linked qualitatively to meridional shifts of the westerly jet circulation. Additionally,*
*the Sofular Cave $\delta^{13}\text{C}$ record from the Asian Westerlies domain is presumably related*
*to the local temperature (Fleitmann et al., 2009; Held et al., 2024), with a negative*
*excursion corresponding to warmer and wetter climate and thus a northward shift of*
*the Westerlies. Besides, during the last glacial period, Greenland ice-core*
*dust-derived $[\text{Ca}^{2+}]$ records primarily reflect the latitudinal position and intensity of*

*the NH westerly winds, as well as the hydroclimate conditions in Asian dust source*
*regions (Dong et al., 2022; Erhardt et al., 2019). These records lend further support*
*to the perception that boreal Westerlies shifted northward (southward) during*
*interstadials (stadials), at least qualitatively.”*

**Comment 7.**

The discussion on the ISM, SASM and EASM is rather throwaway and doesn't
consider the vast body of literature out there. To simply label them as just tropical vs.
sub-tropical dominated is oversimplifying the modern and past climatology.

**Response:** We appreciate the reviewer's comment. Accordingly, we added more
information about the ISM and EASM systems with the modern and past climatology
incorporated. The following paragraphs have been added to the revised version
between Line 73 and Line 77 as well as Line 331 to Line 361:

*"ASM system comprises the Indian summer monsoon (ISM) and East Asian*
*summer monsoon (EASM) subsystems (An et al., 2012; Wang et al., 2017) (Fig. 1d).*
*While the ISM operates primarily as a tropical system driven by low-level*
*southwesterlies, the EASM functions as a subtropical system strongly influenced by*
*the westerly jet stream and Tibetan Plateau (TP) orography (Chiang et al., 2015,*
*2020; Molnar et al., 2010; Schiemann et al., 2009; Xie, 2024) (Methods)."*

*"The different heating of the atmospheric column between continents and oceans*
*drives planetary-scale circulation patterns (Xie, 2024). As for the Asian summer*
*monsoon (ASM), its onset is closely accompanied by a northward shift of the*
*Inter-Tropical Convergence Zone (ITCZ) on a seasonal scale (Cheng et al., 2012;*
*Schneider et al., 2014). The ASM transports moisture and heat from tropical oceans*
*northward to South Asia and East Asia, bringing abundant summer monsoon rainfall*
*(Fig. 1d; Supplementary Fig. 5i). Notably, the ASM contains both dynamic and*
*thermodynamic aspects, and in this study, we mainly focus on dynamics of the ASM*
*circulations.*

*The vast ASM system consists of three major subsystems, the ISM, the EASM and*
*the Western North Pacific summer monsoon (WNPSM) systems (Wang et al., 2017;*
*Wang & Lin, 2002) (Fig. 1d), and this study focuses on the first two subsystems. The*
*ISM is essentially a tropical system (An et al., 2012; Wang et al., 2017; Xie, 2024),*
*characterized by strong low-level southwesterly jet, which extends from the*
*Mascarene High in Southern Hemisphere, across the Arabian Sea and the Bay of*
*Bengal, to the north and northeast of India (Fig. 1d). The deep convections in the*

system set up in mid-May, followed by the onset of the ISM in June. The mid-latitude
westerly jet stream is generally not of first-order importance (Xie, 2024). Indeed, the
classic Gill model, which takes into account the displacement of heat sources around
north of the equator to mimic the summer monsoon circulation (Gill, 1980), captures
the major features of this tropical monsoon system.

The EASM is effectively a subtropical monsoon system, characterized by
low-level southwesterly and southerly moisture and heat transports over the low- and
mid-latitude of East Asia (Fig. 1d). The westerly jet is crucial for the EASM, while the
TP is an important orographic forcing that steers the westerly jet stream (Chiang et
al., 2015, 2020; Molnar et al., 2010; Schiemann et al., 2009). During Spring, the jet
position swings between the north and south of the TP. During early summer, a
decrease in the temperature gradient between high- and low-latitudes pushes the jet
stream to the north of the TP, triggering a northward jump of the EASM. This stage is
characterized by the Meiyu-Baiu rain band extending from the mid- and
lower-reaches of the Yangtze River to Japan (Ding & Chan, 2005). During late-July,
the westerly jet stream jumps further northward, and the Meiyu-Baiu rain belt
vanishes, allowing the southerly-brought moistures to be transported further
northward to the deep interior of the continent (Chiang et al., 2020; Schiemann et al.,
2009)."

**Comment 8.**

There are elements of circulatory in the analysis of the data because they are tuned to
the GICC05 chronology. The methods outline the proxies used, but surely if you are
going to tune a record then comparable proxies should be used, i.e. dust for dust, for
acknowledge that is now an accepted approach.

**Response:** We sincerely appreciate the reviewer's insightful comments, which have
provided us with an opportunity to further clarify our methodology and results
regarding the "tuning" process.

First of all, we would like to emphasize that our analysis of regional temporal phases
is not subject to circular reasoning. This is because that the analysis is essentially
based on a comparison of absolutely-dated Asian and European speleothem records
(Supplementary Fig. 5j, k), rather than relying on the tuned Greenland ice core
records. Furthermore, the primary focus of our study is the comparison of event
amplitudes between short and long interstadials, which remains unaffected by the
"tuning" of ice-core chronology either.

Regarding the chronology alignment, we would like to clarify that we did not
tune our data to the GICC05 chronology. Instead, we aligned the Greenland ice-core

chronology based on GICC05 with our speleothem chronology based on U-Th dating
results. This alignment was achieved through a correlation strategy between the
Greenland ice-core dust-derived $[Ca^{2+}]$ and the ASM speleothem $\delta^{18}O$ records, as
described in Dong et al. (2022).

We concur with the reviewer that, ideally, comparable proxies should be used
when tuning one record to another. For instance, Serno et al. (2015) employed dust
flux records as a chronostratigraphic tool to link the records from the North Pacific
and Greenland. However, applying this approach to the precise comparison of
centennial-millennial events, such as our study here, presents unique challenges.
These challenges arise from the deficiencies in both temporal resolution and
chronological precision of the dust reconstructions outside Greenland and Antarctica,
as well as potential regional divergences.

It is important to note that similar proxies from different locations may carry
distinct environmental implications. While Greenland ice-core $\delta^{18}O$ records primarily
reflect the local temperature and/or sea-ice extent (Buizert et al., 2024; NGRIP Project
Members, 2004; Sime et al., 2019), ASM speleothem $\delta^{18}O$ records represent the
monsoon dynamics, making a direct correlation between them seemingly less
promising. Nevertheless, our strategy for the correlation between Greenland ice-core
dust-derived $[Ca^{2+}]$ records and the ASM speleothem $\delta^{18}O$ records is supported by
several lines of evidence. These records have been shown to vary synchronously
within decadal errors during the late last glacial period (Dong et al., 2022). This
synchronicity can be attributed to the fact that the Taklimakan Desert and the Gobi
Desert in Asia served as the primary dust sources to central Greenland during the last
glacial period (Biscaye et al., 1997; Svensson et al., 2020; Újvári et al., 2022). The
dust transport via the Northern Hemisphere Westerlies is closely coupled with the
large-scale ASM circulation (Cheng et al., 2019; Chiang et al., 2015). As such, the
correlation between Greenland ice core $[Ca^{2+}]$ and ASM speleothem $\delta^{18}O$ records is
causally plausible, which is further substantiated by the extant volcanic evidence and
radiocarbon ages (Dong et al., 2022 and refs. therein), as well as a number of recent
studies (e.g., Muschitiello & López, 2024; Sinnl et al., 2023).

*Comment 9.*

The difference between the records is only an amplitude of 0.5 per mille, given that
you have an isotope enabled model, sure you can quantify the difference in vapor
source to drive this?

**Response:** We sincerely appreciate the reviewer's valuable suggestion regarding the
inclusion of more precise numerical calculation on moisture compositions. While we

acknowledge the potential benefits of such detailed information, we believe that it
may not significantly enhance the overall understanding of the processes under
investigation. This is because our current analysis has already yielded robust results
that provide a solid qualitative understanding of the dynamics involved. Below, we
elaborate further on this perspective to clarify our reasoning.

In the present version, we employ vertical integrated water-vapor transports (WT)
to support that there is increased import of the western Pacific water vapors to the
southeastern China during the overshoot phases (Fig. 4c), which is a result of the
northward shift of the Westerlies associated with the warmer northern high-latitudes
(Fig. 5c, d), in tandem with the northwestern-ward shift of West Pacific subtropical
high (WPSH) (Fig. 5a; Supplementary Fig. 6f). In addition, the available
Westerlies-dominated records (as elaborated in our response to Comments 5 and 6)
and the broad data-model consistency (Line 276–Line 293 in the revised manuscript),
alongside the successful application to explain the different $\delta^{18}\text{O}$ trajectories between
the EASM and ISM during the long interstadials (see response to Comment 1),
provide additional strong support for our conclusions. These lines of evidence
collectively provide qualitative understanding of the governing mechanisms of our
records, although their amplitude difference is relatively small ($\sim 0.5\text{‰}$).

Moreover, the differences in simulated seawater $\delta^{18}\text{O}$ between the overshoot and
interstadial phases offer additional evidence to substantiate our arguments. Generally,
isotopic values in water vapor and surface seawater $\delta^{18}\text{O}$ exhibit an anti-correlation
due to isotopic fractionation during evaporation. Specifically, increased evaporation
driven by warming enriches surface seawater $\delta^{18}\text{O}$ while depleting the $\delta^{18}\text{O}$ of in-situ
water vapor, and vice versa. Consequently, if there were no changes in monsoon
circulation and moisture composition between the overshoot and interstadial phases in
the EASM region, the $\delta^{18}\text{O}$ values of precipitation ($\delta^{18}\text{O}_p$) would become enriched
during interstadial periods, which are characterized by cooler sea surface temperatures
and weaker evaporation in the subtropical western Pacific. However, our experimental
results contradict this expectation, showing instead that $\delta^{18}\text{O}_p$ becomes more depleted
during these periods (note: $\delta^{18}\text{O}_p$ responses in ISM region are in line with our
expectation due to no significant changes in their moisture composition) (Fig. 4c).
Therefore, we attribute this depletion to changes in moisture composition resulting
from shifts in atmospheric circulation.

Quantifying proportions of different sourced water vapors is sensible and
valuable for observations. However, it might not be meaningful based on a single
model output, given its uncertainty and potential biases. We hence suggest a more
comprehensive quantification using multi-model outputs in future studies.

Overall, we believe that the current results are sufficient to advance our
understanding of the mechanisms driving the observed changes. We thank the
reviewer for raising these important points and hope our response clarifies the
rationale behind our approach.

*Comment 10.*

I found difficult to follow your logic with regards to the AMOC overshoot and
interstadial phases. Is the model not sensitive enough to reproduce the short vs long
DOs? It could just be my misunderstanding, but some clarity would be good.

**Response:** According to the suggestion, we have rewritten the Methods section and
introduced more detailed information about the experimental details of our model. In
our simulations, the long interstadials were specifically designed to incorporate the
overshoot phases (Fig. 4a), a feature consistent with observations of most long
interstadials in Greenland. These interstadials typically exhibit a "sawtooth" structure,
characterized by a brief overshoot phase followed by a relatively stable condition, as
documented in NGRIP and GISP2 ice-core $\delta^{18}\text{O}$ records spanning DO-3 to DO-14
(Fig. 3a) (Dansgaard et al., 1993; Grotes et al., 1993; Johnsen et al., 1992;
Rasmussen et al., 2014). This design enables us to effectively analyze the differences
between long and short DO interstadials.

*Comment 11.*

The literature around the land-sea contrast has been significantly updated. It invokes
the Himalayas as a block to westerly flow, so probably important in your thinking
around connections to the North Atlantic. The idea of westerly jet circulation is
nothing new see the Chiang et al., papers about this (included in your ref list).

**Response:** We sincerely thank the reviewer for the insightful comment. We
acknowledge that the literature surrounding land-sea contrast has been substantially
updated, and we agree that the Himalayas play a critical role as a barrier to westerly
flow, which is indeed relevant to our discussion of connections to the North Atlantic.
The importance of the Tibetan Plateau in steering the westerly jet has been addressed
in our response to Comment 7, while the linkage to the North Atlantic is elaborated in
our response to Comment 6.

While the concept of the Westerlies influencing the monsoon climate is
well-established in the literature, this does not imply that the topic has been
exhaustively explored. As highlighted in our responses to Comments 3 and 7, existing
paleoclimate studies have primarily focused on essential contrasts between

millennial-scale interstadials and stadials (e.g., Chiang et al., 2015; Zhang et al.,
2018). However, the role of the westerly jet in distinguishing between long and short
DO interstadials remains poorly understood, largely due to the scarcity of
high-resolution records and pertinent model scrutinization (e.g., with water isotopes).
Our study, leveraging new high-resolution speleothem records and state-of-the-art
isotope-enabled climate modeling, provides novel insights into the nuanced
modulation of East Asian water vapor transport by the boreal Westerlies during short
and long interstadials. This represents a significant advancement in our understanding
of this complex topic.

*Comment 12.*

When discussing the transport of water vapor from the western Pacific to the EASM
region, how much more water vapor would there? Can you calculate this from the
model? What about changes in the isotopic composition of the sea water? Does that
change?

**Response:** Thanks for the suggestion. In the revised version of our manuscript, we
showed the changes in mean annual $\delta^{18}\text{O}_p$ over continents and the mean annual $\delta^{18}\text{O}$
of sea-surface water ($\delta^{18}\text{O}_{sw}$) over oceans (Fig. 4b, c). The results indicate that $\delta^{18}\text{O}_{sw}$
become more depleted during interstadial periods compared with overshoot phases
(Fig. 4c), offering additional evidence to substantiate our arguments (as detailed in
our response to Comment 9).

*Comment 13.*

You state that the water vapor in the ISM is remotely sourced from the Indian Ocean,
are you suggesting it is coming from the Southern Hemisphere? Modern climatology
shows that vapor is very much locally sourced from the west coast of India in the
Arabian Sea and from the Bay of Bengal.

**Response:** Sorry for the misunderstanding caused. We are not suggesting the water
vapor comes from the Southern Hemisphere, and we agree with the reviewer that
vapor is mainly sourced from nearby oceans, as suggested by previous studies.
Modern climatology studies (e.g., Cai et al., 2018) have shown that as for the EASM
domain, the summer monsoon months have increased vapor contribution from remote
sources, such as the Bay of Bengal, compared with pre-monsoon months. Therefore,
we have rephrased the sentence between Line 247 and Line 251 as follow:

*"As a result, more near-source water vapor from the western Pacific with enriched*
*oxygen isotopic values would effectively reduce the magnitude of the isotopic*

depletion in EASM precipitation during the short DO warming (DO-15.1) (Fig. 4b, c),
thus accounting for the contrasting speleothem $\delta^{18}\text{O}$ signals between DO-15.1 and
long DO events."

**Reference:**

An, Z. et al. Global Monsoon Dynamics and Climate Change. *Annu. Rev. Earth Planet. Sci.* **43**,
29-77 (2015).

Biscaye, P. E. et al. Asian provenance of glacial dust (stage 2) in the Greenland Ice Sheet Project 2
Ice Core, Summit, Greenland. *J. Geophys. Res. Oceans* **102**, 26765-26781 (1997).

Buizert, C. et al. The Greenland spatial fingerprint of Dansgaard–Oeschger events in observations
and models. *Proc. Natl. Acad. Sci. U. S. A.* **121**, e2402637121 (2024).

Cai, Y. et al. High-resolution absolute-dated Indian Monsoon record between 53 and 36 ka from
Xiaobailong Cave, southwestern China. *Geology* **34**, 621-624 (2006).

Cai, Z., Tian, L. & Bowen, G. J. Spatial-seasonal patterns reveal large-scale atmospheric controls
on Asian Monsoon precipitation water isotope ratios. *Earth Planet. Sci. Lett* **503**, 158-169
(2018).

Cheng, H., Sinha, A., Wang, X. et al. The Global Paleomonsoon as seen through speleothem
records from Asia and the Americas. *Clim Dyn* **39**, 1045–1062 (2012).

Cheng, H. et al. Chinese stalagmite paleoclimate researches: A review and perspective. *Sci.*
*China-Earth Sci.* **62**, 1489-1513 (2019).

Cheng, H. et al. Orbital-scale Asian summer monsoon variations: Paradox and exploration. *Sci.*
*China-Earth Sci.* **64**, 529-544 (2021a).

Cheng, H. et al. Onset and termination of Heinrich Stadial 4 and the underlying climate dynamics.
*Commun Earth Environ* **2**, 230 (2021b).

Chiang, J. C. H. et al. Role of seasonal transitions and westerly jets in East Asian paleoclimate.
*Quat. Sci. Rev.* **108**, 111-129 (2015).

Chiang, J. C. H., Herman, M. J., Yoshimura, K. & Fung, I. Y. Enriched East Asian oxygen isotope
of precipitation indicates reduced summer seasonality in regional climate and westerlies.
*Proc. Natl. Acad. Sci. USA* **117**, 14745-14750 (2020).

Dansgaard, W. et al. Evidence for general instability of past climate from a 250-kyr ice-core
record. *Nature* **364**, 218-220 (1993).

Ding, Y., Chan, J.C.L. The East Asian summer monsoon: an overview. *Meteorol. Atmos. Phys.* **89**,
117e142. (2005).

Dong, X. et al. Coupled atmosphere-ice-ocean dynamics during Heinrich Stadial 2. *Nat Commun.*
**13**, 5867 (2022).

Erhardt, T. et al. Decadal-scale progression of the onset of Dansgaard–Oeschger warming events.
*Clim. Past* **15**, 811-825 (2019).

Fleitmann, D. et al. Timing and climatic impact of Greenland interstadials recorded in stalagmites

from northern Turkey. *Geophys. Res. Lett.* <https://doi.org/10.1029/2009GL040050> (2009).

Gill A. Some simple solutions for heat-induced tropical circulation. *Quarterly Journal of the*
*Royal Meteorological Society* **106**, 447-462 (1980).

Grootes, P. M. & Stuiver, M. Oxygen 18/16 variability in Greenland snow and ice with 10⁻³- to
10⁵- year time resolution. *J. Geophys. Res.-Oceans* **102**, 26455-26470 (1997).

He, C. et al. Hydroclimate footprint of pan-Asian monsoon water isotope during the last
deglaciation. *Sci. Adv.* **7**, eabe2611 (2021).

Held, F., Cheng, H., Edwards, R.L. et al. Dansgaard-Oeschger cycles of the penultimate and last
glacial period recorded in stalagmites from Türkiye. *Nat Commun* **15**, 1183 (2024).

Hu, J., Emile-Geay, J., Tabor, C., Nusbaumer, J. & Partin, J. Deciphering Oxygen Isotope Records
From Chinese Speleothems With an Isotope-Enabled Climate Model. *Paleoceanogr.*
*Paleoclimatol.* **34**, 2098-2112 (2019).

Johnsen, S. J. et al. The $\delta^{18}\text{O}$ record along the Greenland Ice Core Project deep ice core and the
problem of possible Eemian climatic instability. *J. Geophys. Res.-Oceans* **102** (1997).

Kathayat, G. et al. Interannual oxygen isotope variability in Indian summer monsoon precipitation
reflects changes in moisture sources. *Commun. Earth. Environ.* **2**, 96 (2021).

Liu, X. et al. New insights on Chinese cave $\delta^{18}\text{O}$ records and their paleoclimatic significance.
*Earth-Sci. Rev.* **207**, 103216 (2020).

Molnar, P., Boos, W. R. & Battisti, D. S. Orographic Controls on Climate and Paleoclimate of Asia:
Thermal and Mechanical Roles for the Tibetan Plateau. *Annu. Rev. Earth Planet. Sci.* **38**,
77-102 (2010).

Moseley, G. E. et al. Multi-speleothem record reveals tightly coupled climate between central
Europe and Greenland during Marine Isotope Stage 3. *Geology* **42**, 1043-1046 (2014).

Muschitiello, F. and Aquino-Lopez, M. A.: Continuous synchronization of the Greenland ice-core
and U–Th timescales using probabilistic inversion, *Clim. Past* **20**, 1415–1435 (2024).

Nagashima, K., Tada, R., Tani, A., Sun, Y., Isozaki, Y., Toyoda, S., Hasegawa, H. Millennial-scale
oscillations of the westerly jet path during the last glacial period. *J. Asian Earth Sci.* **40**,
1214e1220 (2011).

Rasmussen, S. O. et al. A stratigraphic framework for abrupt climatic changes during the Last
Glacial period based on three synchronized Greenland ice-core records: refining and
extending the INTIMATE event stratigraphy. *Quat. Sci. Rev.* **106**, 14-28 (2014).

Schiemann, R., Lüthi, D., Schar, C. Seasonality and interannual variability of the westerly jet in
the Tibetan Plateau region. *J. Clim.* **22**, 2940e2957 (2009).

Schneider, T., Bischoff, T. & Haug, G. Migrations and dynamics of the intertropical convergence
zone. *Nature* **513**, 45–53 (2014).

Serno, S., G. Winckler, R. F. Anderson, E. Maier, H. Ren, R. Gersonde, and G. H. Haug.
Comparing dust flux records from the Subarctic North Pacific and Greenland: Implications
for atmospheric transport to Greenland and for the application of dust as a
chronostratigraphic tool, *Paleoceanography* **30**, 583–600 (2015).

Sime, L.C., Hopcroft, P.O. & Rhodes, R.H. Impact of abrupt sea ice loss on Greenland water
isotopes during the last glacial period. *Proc. Natl. Acad. Sci. U. S. A.* **116**: 4099–4104 (2019).

Sinnl, G., Adolphi, F., Christl, M., Welten, K. C., Woodruff, T., Caffee, M., Svensson, A.,
Muscheler, R., and Rasmussen, S. O. Synchronizing ice-core and U/Th timescales in the Last
Glacial Maximum using Hulu Cave ¹⁴C and new ¹⁰Be measurements from Greenland and
Antarctica. *Clim. Past* **19**, 1153–1175 (2023).

Spötl, C., Mangini, A. & Richards, D. A. Chronology and paleoenvironment of Marine Isotope
Stage 3 from two high-elevation speleothems, Austrian Alps. *Quat. Sci. Rev.* **25**, 1127-1136
(2006).

Svensson, A., Biscaye, P. E. & Grousset, F. E. Characterization of late glacial continental dust in
the Greenland Ice Core Project ice core. *J. Geophys. Res.-Atmos.* **105**, 4637-4656 (2000).

Tabor, C. R. et al. Interpreting Precession-Driven $\delta^{18}\text{O}$ Variability in the South Asian Monsoon
Region. *J. Geophys. Res.-Atmos.* **123**, 5927-5946 (2018).

Újvári, G. et al. Greenland Ice Core Record of Last Glacial Dust Sources and Atmospheric
Circulation. *J. Geophys. Res.-Atmos.* **127**, e2022JD036597 (2022).

Wang, B., Lin, H. Rainy season of the Asian-Pacific summer monsoon. *J. Clim.* **15**, 386–396
(2002).

Wang, P. et al. The global monsoon across time scales: Mechanisms and outstanding issues.
*Earth-Sci. Rev.* **174**, 84-121 (2017).

Wang, Y. et al. A High-Resolution Absolute-Dated Late Pleistocene Monsoon Record from Hulu
Cave, China. *Science* **294**, 2345-2348 (2001).

Wu, G. et al. The Influence of Mechanical and Thermal Forcing by the Tibetan Plateau on Asian
Climate. *Journal of hydrometeorology-special section* **8**, 770-788 (2007).

Xie, S. Chapter 5 - Summer monsoons. Coupled atmosphere-ocean dynamics. San Diego: Elsevier,
101-138 (2024).

Zhang, H. et al. East Asian hydroclimate modulated by the position of the westerlies during
Termination I. *Science* **362**, 580-583 (2018).

Zhang, H. et al. A data-model comparison pinpoints Holocene spatiotemporal pattern of East
Asian summer monsoon. *Quat. Sci. Rev.* **261**, 106911 (2021).

Zhao, J. et al. Orchestrated decline of Asian summer monsoon and Atlantic meridional overturning
circulation in global warming period. *The Innovation Geosci.* **1**, 100011,
doi:10.59717/j.xinn-geo.2023.100011 (2023).

A point-by-point response

(Original comments are in *blue*, and our responses are in *black*)

1. Comments from Reviewer #1:

General remark.

I appreciate the authors' effort in revising the manuscript. My previous concerns have been well addressed. I have no further comments.

Response: We sincerely thank the reviewer for his/her constructive comments, which have considerably improved our manuscript.

2. Comments from Reviewer #2:

General remark.

This manuscript presents a very high-resolution $\delta^{18}\text{O}$ record from four stalagmites in the Indian and East Asian monsoon regions, covering a brief interstadial period with precise age control. I agree that this dataset is valuable for deepening our understanding of spatial hydrological variability during interstadials.

However, in the revised manuscript, the central scientific questions remain unclear. For the study to engage a broader readership of this journal, the authors should state more clearly their key motivations and implications of their work:

Response: We thank the reviewer for his/her constructive feedback. In the revised version, we have explicitly stated our key motivations and the broader significance of our findings thereby making our results more accessible to a broad, interdisciplinary readership. The key changes in the manuscript are summarized here as follow. First, we emphasize why understanding tropical and subtropical hydroclimate responses to interstadial events is essential for interpreting global climate teleconnections. Second, we clarified our interstadial classification framework based on duration and mechanistic differences. Third, we highlight that short interstadials induce a much further northward shift in the Westerlies, affecting precipitation isotopic values in East Asia via enhanced vapor transport from near-source regions (western North Pacific), a governing process distinct from those in Indian monsoon and those involved on interannual timescale (Chiang et al., 2020). Finally, we refined Fig. 4d and its caption to more clearly illustrate this mechanism. Collectively, these revisions articulate how this study fill a critical gap in understanding variations of midlatitude – tropical interactions between different interstadial states, and how high-latitude or polar warming modulates both Northern Westerlies position and EASM hydroclimate in ways not previously recognized.

*Comment 1.*

Why is it important to investigate climate variability in tropical and subtropical regions
during the interstadials? Simply citing the lack of high-resolution data may not be
sufficient.

**Response:** Tropical and subtropical regions as important heat and moisture sources
play a critical role in inter- and intra-hemisphere climate teleconnections and have
experienced evident changes during past abrupt climate changes, i.e. Dansgaard-
Oeschger (DO) events (e.g., Cheng et al., 2019, 2022; Corrick et al., 2020; Dong et al.,
2022; Wang et al., 2017). Nonetheless, most process-understanding studies regarding
DO events have focused on the northern and southern high latitudes and on those
durations more than one thousand years (i.e. long DO events defined in this study) (e.g.,
Buizert et al., 2018, 2024; Menviel et al., 2020; WAIS Divide Project Members, 2015).
This apparently hampers a complete understanding of DOs' impacts and associated
mechanisms. This study fills this gap of knowledge – we corroborate two types of DO
events based on new high-resolution speleothem $\delta^{18}\text{O}$ records from EASM region and
elaborate mechanisms of their distinct $\delta^{18}\text{O}$ changes applying an isotope-enabled
climate model. In particular, the new mechanism advances our understanding of how
midlatitude-tropical interactions shape EASM hydroclimate variability on millennial
timescales, as elaborated in the following:

Previously, seasonal migration of Northern Westerlies has been proposed to explain
interannual variations in precipitation isotopic values in EASM region, owing to its
modulation on the $\delta^{18}\text{O}_p$ transition from isotopically heavier winter to the lighter
summer regime and hence the mean annual $\delta^{18}\text{O}_p$ values (Chiang et al., 2020). The
authors infer that this mechanism may also work for millennial-scale $\delta^{18}\text{O}_p$ variability,
as marine and lacustrine records have documented large-scale shifts in the Northern
Hemisphere Westerlies (e.g., An et al., 2012; Nagashima et al., 2011). However, no
conclusive and direct lines of evidence have been presented to test this hypothesis. Our
results fill this gap and corroborate the applicability of this mechanism for millennial-
scale EASM $\delta^{18}\text{O}_p$ variability. In addition, we further identify a non-trivial effect of the
Westerlies shift during interstadials. That is, a nuanced northward shift of the summer
Westerlies, which links to additional Northern Hemisphere high-latitude warming
during short interstadials or during the AMOC overshoot phase of long interstadials,
enhances the proportion of near-sourced vapor from the western North Pacific,
enriching mean annual isotopic values of precipitation in eastern China (the region of
interest in this study). Most importantly, only combining the two effects of the
Westerlies shift together can we explain the observed $\delta^{18}\text{O}$ changes in EASM region in
different DO events. Therefore, our results substantially advance our understanding of

the mechanisms behind millennial-scale EASM $\delta^{18}\text{O}_p$ variability, providing a more
complete dynamical framework that explains the co-variability of the AMOC and
EASM across timescales from interannual to millennial (See details in our response to
Comment 3).

*Comment 2.*

In what context do the authors classify interstadials based on their duration and try to
find the difference among them? Could the results of this study offer new insights into
the length of interstadials and their underlying mechanisms?

**Response:**

Our classification of interstadials draws on Greenland ice-core $\delta^{18}\text{O}$ records
(NGRIP/GISP2), which clearly differentiate between short (e.g., DO-15.1) and long
interstadials based on duration and isotopic structure. Short interstadials and the initial
“overshoot” phases of long interstadials are marked by abrupt, high-amplitude warming
followed by cooling. These phases likely reflect a transient phase when excess
subsurface ocean heat accumulated during stadials is released (e.g., Liu et al., 2009; Su
et al., 2016). In contrast, long interstadials further include a stable active AMOC phase,
where persistent northward oceanic heat transport sustained a stable warm condition
without additional subsurface heat release. We have incorporated these important
contextual points in our revised manuscript.

Our findings offer insights into how these different interstadial modes modulate
subtropical hydroclimates, particularly via shifts in the Westerlies and associated vapor
transport pathways. Our records do not directly resolve the mechanisms governing
interstadial length, underscoring a need for future studies for constraining the interplay
among AMOC variability, ice-sheet dynamics, ocean heat storage and perhaps tropical
monsoon feedbacks that ultimately shape the duration and structure of North Atlantic
interstadials.

*Comment 3.*

Why is the result of a northward westerly jet position during this short interstadial
notable, especially when northward patterns have been previously reported for longer
interstadials? I think that clarifying these scientific motivations and their implications
would strengthen the manuscript.

**Response:** We thank the reviewer for raising this important point. While previous
studies have indeed documented northward shift during long interstadials, our results
reveal several novel aspects that clarify their broader dynamical significance.

First, the northward shift during short interstadials is more rapid and pronounced
than during the equilibrium phases of longer interstadials. The short interstadials
represent transient warming periods (refer to our response to Comment 2). This reflects
the steep meridional temperature gradient driven by abrupt high-latitude warming and
oceanic heat release during the AMOC transient phases. This dynamic response has not
been previously quantified or mechanistically linked to isotopic changes.

Second, we show that this mechanism also works in the initial overshoot phases of
long interstadials. Consistent with our prediction that amplified high-latitude warming
leads to less depleted $\delta^{18}\text{O}$ in Southeast China, the Xianyun Cave record shows
systematically higher $\delta^{18}\text{O}$ values during the overshoot phases than in the equilibrium
phases of DO-8, -12, and -14 (Fig. 3c). In contrast, the Bittoo Cave record (ISM region)
shows no such phase-dependent isotopic change (Fig. 3d). This regional contrast
validates our interpretation and highlights the influence of westerly dynamics on EASM
hydroclimate. Crucially, these results also underscore that the intensity of high-latitude
warming rather than interstadial duration is the primary driver of how far the Westerlies'
shifts northwards.

Third, our findings provide the first clear evidence that the Westerlies' position is
not simply binary (north during all interstadials versus south during stadials), but rather
exhibits a continuum of responses depending on the degree of high-latitude warming.
This continuum reflects the underlying ocean-atmosphere dynamics - from the intense
but short-lived heat release during overshoots to the more sustained but moderate
forcing during equilibrium phases.

Finally, although previous work (e.g., Chiang et al., 2020) suggested the
mechanism accounting for interannual westerly- $\delta^{18}\text{O}$ relationships may be applicable
to the millennial timescales; our isotope-enabled modeling results, combined with new
speleothem data, test this hypothesis and reveal a complementary but equally important
mechanism to understand millennial-scale EASM $\delta^{18}\text{O}_p$ variability (see our response to
Comment 1). Fig. 4d highlights the distinct isotopic imprint in Southeast China,
providing direct evidence for a mechanistic connection between Westerlies and
precipitation $\delta^{18}\text{O}$. Together, these findings clarify how the intensity of high-latitude
warming and not just event duration governs westerly behavior and downstream
monsoon hydroclimate. We have incorporated these important contextual points in our
revised manuscript to better highlight how our work advances beyond previous findings
and fills critical research gaps in paleoclimate dynamics.

Comment 4.

Lines 68-70: Are there any other short interstadials, around 100 years in duration,

besides DO-15.1?

**Response:** Based on Greenland ice-core $\delta^{18}\text{O}$ records, there are other short interstadials
in Marine Isotope Stage 4 (MIS4), e.g., GI-16.2 (120-year duration) and GI-17.2 (140-
151 year duration) (Rasmussen et al., 2014). To date, these short interstadials remain poorly
constrained in low-latitude paleoclimate records, warranting further investigation.

*Comment 5.*

Lines 84-91: The significance of addressing these knowledge gaps is not sufficiently
explained.

**Response:** Thanks for the reviewer. We have rewritten the introduction section in the
revised version of our manuscript to highlight the significance of addressing the current
knowledge gaps (please refer to our responses to Comment 1 and 3).

*Comment 6.*

Lines 102-106: The current description may cause misunderstanding. It might be
already established that the westerlies shift northward during interstadials compared to
stadials. Therefore, the authors should clarify that what they observe during the short
interstadial is a further northward shift compared to the shift observed during longer
interstadials.

**Response:** We have clarified this in the revised version of our manuscript (Line 128-
130).

*Comment 7.*

Lines 216-225: I still wonder why the AMOC overshoot at the onset of interstadials is
considered an appropriate analogue for short interstadials, simply because of the similar
$\delta^{18}\text{O}$ values observed during the overshoot phase and during D–O 15.1 in both
Greenland ice cores and stalagmite records from the EASM region. Are the mechanisms
that generate the overshoot and those responsible for the short interstadial comparable?

**Response:** The justification for comparing short interstadials with the overshoot phase
of long interstadials lies in their shared dynamical features. Both represent transient
responses to abrupt high-latitude warming, driven by the rapid release of heat from
subsurface ocean reservoirs following stadials (e.g., Liu et al., 2009; Su et al., 2016).
This results in a pronounced but short-lived warming signal, followed by stable warm
conditions—captured as the characteristic “sawtooth” $\delta^{18}\text{O}$ pattern in Greenland ice
cores and replicated in our model experiment (Fig. 3a, b). This mechanistic analogy is
further supported by the consistency of regional $\delta^{18}\text{O}$ signals: the gradual evolution

seen in southeastern Chinese speleothems during the overshoot phases of long
interstadials contrast with the more abrupt shifts in Indian records (See response to
Comment 3). These patterns validate the use of the overshoot phase as a process
analogue for short interstadials.

Lastly, in order to explicitly test whether the overshoot phase affects our
conclusions, we conducted additional analyses by defining interstadial phases
excluding overshoot periods (Supplementary Fig. 7a; Methods). The results remain
consistent with our original findings, showing minimal difference in India and
pronounced differences in Southeast China between the long and short interstadials (Fig.
4d; Supplementary Fig. 7c). This contrast is primarily driven by warmer high-latitude
conditions during the short interstadials (Fig. 5d; Supplementary Fig. 7e), which favors
a further northward shift of the interstadial Northern Westerlies (Fig. 5c, d;
Supplementary Fig. 7d). This facilitates import of isotopically-enriched near-source
water vapor from subtropical western Pacific to Southeast China during the short
interstadials (Fig. 4b, c; Supplementary Fig. 7b). These results further strengthen the
reliability of our conclusions.

Comment 8.

Figure 4d: I thought that many readers, including myself, may find it difficult to grasp
the meaning of the ratio ' $\delta^{18}O_p$ (Interstadial minus Overshoot phases) / (Interstadial
minus Stadial phases)'. Could the authors consider presenting this trend in a more
straightforward way to better illustrate the $\delta^{18}O$ evolution during the overshoot phase?

**Response:** We have changed the sub-title in Fig. 4d to “Impacts of Northern Westerlies
on interstadial ASM $\delta^{18}O_p$ variations”. We have modified the figure caption to allow
readers to easily understand the figure: “*In the core ASM domain, positive values denote
regions with influences of the Westerlies shifts. The values of the ratio approaching 1
indicate a muted $\delta^{18}O_p$ depletion during transitions from stadial to short interstadials
and to overshoot phases of long interstadials, as a consequence of a further northward
shift in Northern Westerlies. Ratios near zero indicate $\delta^{18}O_p$ differences between
interstadials and stadials are independent of the Westerlies shift.*”

**Reference:**

An, Z. et al. Global Monsoon Dynamics and Climate Change. *Annu. Rev. Earth Planet. Sci.* **43**, 29-
77 (2015).

Böhm, E., Lippold, J., Gutjahr, M. et al. Strong and deep Atlantic meridional overturning circulation
during the last glacial cycle. *Nature* **517**, 73–76 (2015).

Buizert, C., Sigl, M., Severi, M. et al. Abrupt ice-age shifts in southern westerly winds and Antarctic

climate forced from the north. *Nature* **563**, 681–685 (2018).

Buizert, C. et al. The Greenland spatial fingerprint of Dansgaard–Oeschger events in observations
and models. *Proc. Natl. Acad. Sci. U. S. A.* **121**, e2402637121 (2024).

Cheng, H. et al. Chinese stalagmite paleoclimate researches: A review and perspective. *Sci. China-
Earth Sci.* **62**, 1489-1513 (2019).

Cheng, H. et al. Milankovitch theory and monsoon. *The Innovation* **3**, 100338 (2022).

Chiang, J. C. H., Herman, M. J., Yoshimura, K. & Fung, I. Y. Enriched East Asian oxygen isotope
of precipitation indicates reduced summer seasonality in regional climate and westerlies. *Proc.
Natl. Acad. Sci. USA* **117**, 14745-14750 (2020).

Corrick, E. C. et al. Synchronous timing of abrupt climate changes during the last glacial period.
*Science* **369**, 963-969 (2020).

Dong, X. et al. Coupled atmosphere-ice-ocean dynamics during Heinrich Stadial 2. *Nat Commun*
**13**, 5867 (2022).

IPCC, 2021: Summary for Policymakers. In: Climate Change 2021: The Physical Science Basis.
Contribution of Working Group I to the Sixth Assessment Report of the Intergovernmental
Panel on Climate Change [Masson-Delmotte, V., P. Zhai, A. Pirani, S. L. Connors, C. Péan, S
Berger, N. Caud, Y. Chen, L. Goldfarb, M. I. Gomis, M. Huang, K. Leitzell, E. Lonnoy, J.B.R.
Matthews, T. K. Maycock, T. Waterfield, O. Yelekçi, R. Yu and B. Zhou (eds.)]. Cambridge
University Press.

Liu, Z. et al. Transient Simulation of Last Deglaciation with a New Mechanism for Bølling-Allerød
Warming. *Science* **325**, 310-314 (2009).

Menviel, L.C. et al. An ice-climate oscillatory framework for Dansgaard–Oeschger cycles. *Nat. Rev.
Earth. Environ.* **1**, 677–693 (2020).

Nagashima, K., Tada, R., Tani, A., Sun, Y., Isozaki, Y., Toyoda, S., Hasegawa, H. Millennial-scale
oscillations of the westerly jet path during the last glacial period. *J. Asian Earth Sci.* **40**,
1214e1220 (2011).

Rasmussen, S. O. et al. A stratigraphic framework for abrupt climatic changes during the Last
Glacial period based on three synchronized Greenland ice-core records: refining and extending
the INTIMATE event stratigraphy. *Quat. Sci. Rev.* **106**, 14-28 (2014).

Su, Z., Ingersoll, A. P. & He, F. On the Abruptness of Bølling–Allerød warming. *J. Climate* **29**,
4965-4975 (2016).

Wang, X. et al. Hydroclimate changes across the Amazon lowlands over the past 45,000 years.
*Nature* **541**, 204-207 (2017).

WAIS Divide Project Members. Precise inter-polar phasing of abrupt climate change during the last
ice age. *Nature* **520**, 661-665 (2015).

A point-by-point response

(Original comments are in *blue*, and our responses are in *black*)

1. Comments from Reviewer #3:

General remark.

The authors have carefully revised the manuscript and provided clear responses to the reviewers' comments.

I believe the revised version meets the journal's standards, and I have no further concerns.

I support the publication of this manuscript.

Response: We sincerely thank the reviewer for his/her constructive comments, which have considerably improved our manuscript.